# L-MTP: Leap Multi-Token Prediction Beyond Adjacent Context for Large Language Models

Xiaohao Liu[1]    Xiaobo Xia[1*]    Weixiang Zhao[2]    Manyi Zhang[3]
Xianzhi Yu[3]    Xiu Su[4]    Shuo Yang[5]    See-Kiong Ng[1]    Tat-Seng Chua[1]
[1]National University of Singapore    [2]Harbin Institute of Technology
[3]Huawei Noah's Ark Lab    [4]Central South University
[5]Harbin Institute of Technology, Shenzhen
{XIAOHAO.LIU@U.NUS.EDU, XIAOBOXIA.UNI@GMAIL.COM}

## Abstract

Large language models (LLMs) have achieved notable progress. Despite their success, next-token prediction (NTP), the dominant method for LLM training and inference, is constrained in both contextual coverage and inference efficiency due to its inherently sequential process. To overcome these challenges, we propose leap multi-token prediction (L-MTP), an innovative token prediction method that extends the capabilities of multi-token prediction (MTP) by introducing a leap-based mechanism. Unlike conventional MTP, which generates multiple tokens at adjacent positions, L-MTP strategically skips over intermediate tokens, predicting non-sequential ones in a single forward pass. This structured leap not only enhances the model's ability to capture long-range dependencies but also enables a decoding strategy specially optimized for non-sequential leap token generation, effectively accelerating inference. We theoretically demonstrate the benefit of L-MTP in improving inference efficiency. Experiments across diverse benchmarks validate its merit in boosting both LLM performance and inference speed. The source code is available at https://github.com/Xiaohao-Liu/L-MTP.

## 1   Introduction

Large language models (LLMs) have demonstrated rapid and remarkable progress, driven by data, computing, and architectural innovation advances [1, 2, 3, 4, 5, 6, 7, 8, 9, 10, 11]. They exhibit strong capabilities in world knowledge acquisition [12, 13] and enable breakthroughs across a wide range of research domains, such as chemistry [14, 15], biology [16, 17], medicine [18, 19], and personalization [20, 21, 22, 23]. As model scales and training data continue to increase, LLMs are attaining ever more powerful generalization and reasoning abilities [24, 25, 26, 27, 28, 29, 30].

Next-token prediction (NTP) remains the mainstream strategy for both training and inference in LLMs [31, 32, 33, 34, 35, 36, 37]. It generates tokens in an autoregressive manner, where each token is predicted based only on the preceding context (see Figure 1(a)). However, despite its conceptual simplicity, NTP results in inefficient generation, and limits the model to a focused yet short contextual horizon, and overlooks "hard" decisions [38, 39]. Intriguingly, LLMs are verified with inherent *pre-planning* capabilities, which indicates the potential of extending NTP to predict multiple tokens at once [39]. This gives rise to the multi-token prediction (MTP) paradigm [39] (see Figure 1(b)). Specifically, by incorporating additional language model heads, MTP enables the parallel prediction of a sequence of adjacent tokens and brings two key benefits. First, it provides a *broader* training

---

[*]Corresponding author.

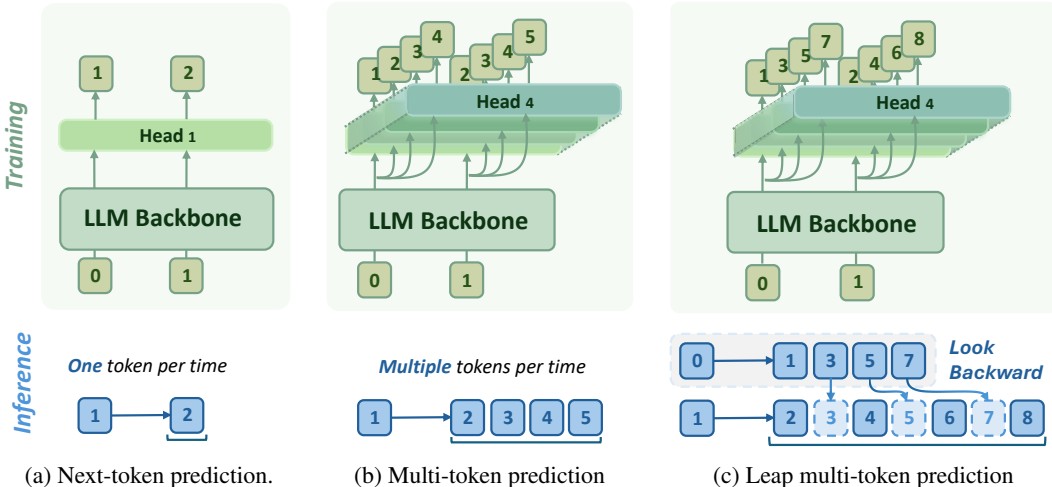

(a) Next-token prediction.    (b) Multi-token prediction    (c) Leap multi-token prediction

Figure 1: **Illustrations of LLM architectures with three prediction paradigms, including NTP (a), MTP (b), and L-MTP (c).** NTP utilizes a single output head for sequential token prediction. MTP employs multiple output heads for adjacent multi-token forecasting. As a comparison, L-MTP reassigns prediction heads to leaping positions. For instance, given 4 heads, L-MTP predicts $[1, 3, 5, 7]$ tokens instead of the adjacent sequence $[1, 2, 3, 4]$ in MTP with a stride of 2 and the initial input token. The top depicts the training difference, while the bottom showcases the inference[2].

signal by supervising multiple upcoming tokens at each step, which can enhance performance in tasks requiring long-range reasoning or planning [39, 40, 41]. Second, it enables *faster* inference by generating multiple tokens in a single forward pass, reducing latency and increasing throughput in applications with efficiency constraints [39].

In this paper, motivated by the philosophy of going broader and faster, we extend the MTP paradigm and propose *leap multi-token prediction (L-MTP)*, which further amplifies both contextual coverage and inference efficiency of LLMs. As illustrated in Figure 1(c), L-MTP introduces a leaping mechanism that skips intermediate tokens and directly predicts non-adjacent future tokens. Structurally, L-MTP retains the core architecture of MTP, where multiple prediction heads are applied in parallel. Nevertheless, instead of targeting consecutive positions, each head in L-MTP is reassigned to predict tokens at leaping intervals (*e.g.*, positions 1, 3, 5, and 7). This yields a broader training signal than MTP, as the model learns to capture longer-range dependencies beyond adjacent-token contexts. During inference, L-MTP further improves generation speed by reusing overlapping context across decoding steps. By jointly predicting multiple and strategically spaced tokens, L-MTP enables each forward pass to generate more tokens per step, which helps reduce the total number of decoding iterations required. This leads to faster inference compared to standard MTP, while maintaining consistency in the generated outputs.

The study of L-MTP can be justified from both human thinking and recent trends in language model reasoning. In human thinking, we rarely reason in a strictly sequential fashion. Instead, we often skip over intermediate elements to complete reasoning more efficiently [42, 43, 44]. This leap-wise reasoning aligns naturally with L-MTP's mechanism of skipping intermediate tokens and predicting non-adjacent ones. Similarly, in language model reasoning, recent advances in efficient reasoning have revealed that many intermediate reasoning steps can be compressed or abstracted without loss of correctness [45, 46, 47]. By predicting tokens at leaping positions, L-MTP mimics this abstraction process, not by explicitly modeling token importance, but by altering the prediction pattern to skip intermediate positions, to accelerate LLM inference.

We provide a theoretical analysis to demonstrate the inference acceleration of our L-MTP, by focusing on the attenuation and consistency of output token probabilities. Besides, through comprehensive experiments, we show that L-MTP can improve the performance of LLMs in a series of tasks

---

[2]To avoid dense or confusing presentation, we here omit prediction, verification, and acceptance sub-procedures during the inference of MTP and L-MTP. More details can be checked in Section 2.

(*e.g.*, math and code tasks) and meanwhile improve inference speed. For instance, L-MTP achieves competitive performance and outperforms MTP at most tasks. L-MTP also boosts existing MTP models with 22% more inference speed-up with the same number of heads. Furthermore, we provide experimental evidence that L-MTP is extendable to speculative decoding techniques, making models up to 4 times faster at inference time across a wide range of settings.

## 2 Preliminaries

In this section, we formulate and detail the training and inference procedures of next-token prediction (NTP) and multi-token prediction (MTP) for large language models (LLMs).

**NTP.** Given input tokens $x_{\leq t}$, where $\leq t$ in the subscript represents the abbreviation of $\{1, 2, \ldots, t\}$, the LLM with parameters $\theta$ predicts the next token $x_{t+1}$ and is optimized via the following objective:

$$\mathcal{L}_{\text{NTP}} = -\sum_T \log p(x_{t+1}|x_{\leq t}; \theta), \tag{1}$$

where $T$ denotes the number of tokens. The decoding process of NTP in inference also follows an autoregressive manner, which generates tokens one by one. The next token is sampled from $p(x_{t+1}|x_{\leq t}; \theta)$.

**MTP.** A natural extension for NTP is MTP [39] that predicts multiple tokens at once. Given input tokens $x_{\leq t}$, the LLM with parameters $\bar{\theta}$, predicts the following $n$ tokens, by involving more output heads (*e.g.*, 4 output heads totally). Therefore, the optimization objective is derived from Eq. (1) to:

$$\mathcal{L}_{\text{MTP}} = -\sum_T \log p(x_{[t+n,\ldots,t+2,t+1]}|x_{\leq t}; \bar{\theta}). \tag{2}$$

In this case, the LLM is requested to pre-plan the adjacent context rather than the single next token. For MTP, during decoding, the next $n$ tokens can be sampled independently, following $p(x_{t+i}|x_{\leq t}; \bar{\theta}), i \in \{1, 2, \ldots, n\}$. Recent research [39, 48] disentangles the LLM with the LLM backbone and output heads, where the former yields hidden states and the latter maps them into a vocabulary distribution. Therefore, the LLM backbone can be equipped with multiple heads to predict the tokens independently according to the shared hidden states. To this end, we have

$$p(x_{t+n,\ldots,t+2,t+1}|x_{\leq t}; \bar{\theta}) = \prod_{i=1}^{n} \Big( p(x_{t+i}|\mathbf{z}_{\leq t}; \theta^i) \cdot p(\mathbf{z}_{\leq t}|x_{\leq t}; \theta') \Big), \tag{3}$$

where $\mathbf{z}$ denotes the hidden states, $\theta'$ represents the parameters for the LLM backbone, and $\theta^i$ is the parameters for the $i$-th output head. Following token predictions, MTP typically incorporates verification and acceptance sub-procedures to determine which tokens are eligible for output (*cf.*, Figure 2). Specifically, the verification sub-procedure invokes LLMs to evaluate the predictions in parallel and records their probabilities for determining the acceptance. Accepted token will be used to update the KV cache, and the process proceeds to the next iteration. Such decoding is proven as a lossless LLM acceleration technique, where the sampling is consistent with the distribution of vanilla autoregressive decoding [49].

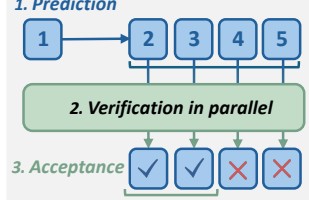

Figure 2: **MTP with self-speculative decoding**, involving three sub-procedures.

## 3 L-MTP: Leap Multi-Token Prediction

**Overview.** Despite the potential of MTP, we go beyond it with one innovative solution to achieve a broader prediction range and faster inference. Unlike conventional MTP, which focuses on consecutive tokens, L-MTP introduces a leap-based strategy, allowing it to predict tokens at non-sequential positions within the context window. This design enables the model to efficiently capture long-range dependencies without the need for dense token predictions. During inference, L-MTP reuses partially overlapping context across prediction steps, maximizing information utilization while minimizing redundant calculations. Given input tokens $x_{\leq t}$, L-MTP aims to predict a sequence of tokens

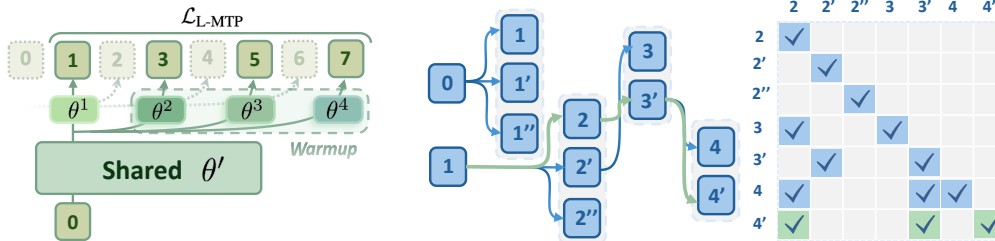

Figure 3: **Training recipe for L-MTP with objective $\mathcal{L}_{\text{L-MTP}}$.** We warm up additional heads $\{\theta^i\}_{i>1}$, and then optimize the whole model, with multiple leap tokens as supervision.

Figure 4: **Incorporation with tree-attention.** We take multiple candidates concurrently to construct the tree paths (*left*), thus exploring the accepted one. Our backward decoding strategy offers consecutive sequences, which can be verified with crafted tree-attention (*right*).

at leaping intervals, specifically at positions, *i.e.*, $x_{[t+k(n-1)+1,...,t+k+1,t+1]}$, where $k$ denotes the number of jumped tokens. For example, with $t$ input tokens and $k=2$, the model is expected to predict the tokens at positions $[t+1, t+3, t+5, \ldots, t+2n-1]$, effectively skipping intermediate tokens in each prediction step. We detail our L-MTP below.

### 3.1 L-MTP Training Recipe

We equip an LLM with multiple output heads for predicting tokens at different positions. The head is a multilayer perceptron (MLP) with the last layer transforming hidden states to vocabulary (see more implementation details in Appendix B.5). We utilize two stages to train the LLM with multiple heads: (1) head warm-up; (2) full model tuning.

**Head warm-up.** We first construct the self-distillation data by inputting the questions while collecting the output from the untapped LLM. These outputs follow the original distribution of LLM's predictions. We optimize new heads by assigning them different supervisions, adhering to the leaping pattern $x_{t+k(n-1)+1}$. The primary goal of this stage is to adapt new heads to the LLM. Therefore, the original head and LLM backbone are frozen. The training objective is formulated as

$$\mathcal{L}^{(1)}_{\text{L-MTP}} = -\sum_T \log p(x_{[t+k(n-1)+1,...,t+k+1]}|\mathbf{z}_{\leq t}; \{\theta^i\}_{i>1}). \tag{4}$$

**Full model tuning.** After that, we use the curated data to continue training the model. At this stage, all the components in our specialized LLM, including the LLM backbone and output heads, are optimized. The optimization objective is defined as

$$\mathcal{L}^{(2)}_{\text{L-MTP}} = -\sum_T \log p(x_{t+1}|x_{\leq t}; \theta', \theta^1) + \beta \cdot \log p(x_{[t+k(n-1)+1,...,t+k+1]}|x_{\leq t}; \theta', \{\theta^i\}_{i>1}), \tag{5}$$

where $\beta$ controls the contribution of additional heads.

### 3.2 L-MTP Inference Procedure

Despite L-MTP offering a broader prediction range, at every pass, it can only predict an incomplete sequence. Fortunately, we can step backward to leverage the prior predictions, or forward to utilize the posterior prediction to compensate for the incompleteness (an alternative solution explained in Appendix C.1). Furthermore, by exploiting speculative decoding techniques (*i.e.*, parallel tree decoding), L-MTP can achieve a larger accept rate, thus achieving further inference accelerate.

**Looking backward.** Given $x_{\leq t}$ tokens[3], L-MTP predicts tokens $\{x_{t+k(i-1)+1}\}_{i\in[n]}$, while leaving gaps between them (*i.e.*, $k-1$ tokens are skipped). However, if we look *backward*, the desired tokens have already been predicted by prior steps. For instance, tokens $\{x_{t+k(i-1)}\}_{i\in[n]}$ are predicted given

---

[3]Typically, $t>1$ with the input containing at least one start token, like "<|begin_of_text|>" in Llama series.

$x_{\leq t-1}$. In this case, we have:

$$\left\{ p(x_{t+i}|x_{\leq t-(i-1) \bmod k})|i \in \{1, 2, \ldots, k(n-1)+1\} \right\}. \tag{6}$$

The continuous token sequence is sampled by looking backwards $(-)$ $k-1$ steps. Here, $\bmod k$ helps to switch the conditions. Since the priors are generated beforehand (or in parallel), we do not need to infer again, but retrieve them.

**Combining with tree attention.** L-MTP seamlessly integrates with speculative decoding by sampling consecutive token sequences for verification. Drawing inspiration from parallel decoding [50, 51, 48, 52], we combine L-MTP with tree attention to enable efficient decoding. We construct a hierarchical tree structure, where the $i$-th layer represents candidate tokens generated by the $i$-th prediction head. Paths in the tree are explored to identify the accepted one. To facilitate parallel verification, we design a tree attention mask that restricts each hidden state to attend only to its ancestors [50, 48]. Figure 4 illustrates the implementation of the tree attention with L-MTP decoding. Further details are provided in Appendix C.2.

## 4 Theoretical Analyses

Given the input $x_{\leq t}$, LLMs are capable of predicting further future tokens, such as $x_{t+i}, i > 1$. This motivates the development of MTP, where future tokens can be predicted from far previous tokens, rather than merely the last ones. By observing the acceptance rates of generated multiple tokens, we draw two properties:

**Definition 1** (Attenuation). *For a language model predicting multiple tokens conditioned on $x_{\leq t}$, the marginal probability of predicting each subsequent token decreases as the prediction horizon increases. Formally, $p(x_{t+1}|x_{\leq t}) > p(x_{t+2}|x_{\leq t}) > \cdots > p(x_{t+n}|x_{\leq t}), \forall i \in \{1, 2, \ldots, n\}$, where $n$ is the maximum prediction horizon (the number of heads), assuming the probabilities are well-defined and non-zero.*

**Remark.** Attenuation reflects the increasing uncertainty in the language model's predictions as it forecasts tokens further into the future. This behavior arises because the prediction of $x_{t+i}$ relies on the fixed context $x_{\leq t}$, and the influence of this context diminishes with increasing $i$. As a result, the model's confidence, as measured by the marginal probability $p(x_{t+i}|x_{\leq t})$, decreases monotonically.

**Assumption 2** (Consistency). *The expected marginal probability of predicting $x_{t+i}$ is stable across arbitrary inputs and follows a predictable function of the prediction horizon $i$. Formally: $\mathbb{E}_{x_{\leq t} \sim \mathcal{D}}[p(x_{t+i}|x_{\leq t})] = f(i), \forall i \in \{1, 2, \ldots, n\}$, where $f(i)$ is a function characterizing the expected probability of $x_{t+i}$.*

**Remark.** Consistency ensures that the language model's predictions for future tokens $x_{t+i}$ exhibit stable statistical behavior in expectation, regardless of the variability in the input sequences $x_{\leq t}$. The function $f(i)$ encapsulates the expected confidence in predicting the $i$-th token ahead, which may decrease with $i$ in accordance with the Attenuation definition (*i.e.*, $f(i) > f(i+1)$).

**Acceptance length.** The expected length for accepted tokens can be expressed as $\mathbb{E}[L] = \sum_{m=1}^{n} p(L > m)$. According to different prediction strategies, we have the following expectations:

$$\begin{cases} \mathbb{E}[L] & = & \sum_{m=1}^{n} \left( \prod_{i=1}^{m} \mathbb{E}_{x_{\leq t} \sim \mathcal{D}}[p(x_{t+i}|x_{\leq t})] \right) \quad \text{(Vanilla strategy)} \\ \mathbb{E}[L]_l & = & \sum_{m=1}^{k(n-1)+1} \left( \prod_{i=1}^{m} \mathbb{E}_{x_{\leq t} \sim \mathcal{D}}[p(x_{t+i}|x_{\leq t-(i-1) \bmod k})] \right) \quad \text{(L-MTP strategy)} \end{cases} \tag{7}$$

where vanilla strategy predicts tokens $x_{t+1}, x_{t+2}, \ldots, x_{t+n}$ sequentially using the hidden state at $t$. L-MTP predicts two interleaved sequences. Specifically, L-MTP uses the hidden state at $t-1$ to compensate for the non-predicted tokens.

**Theorem 3** (Less attenuation, more speed-up). *Let $\gamma$ represent the attenuation coefficient, and $f(i) := \exp[-\gamma \cdot (i-1)]$ be the probability decay function modeling the predictive confidence at step $i$. Then there exists a constant $C > 0$ such that $\mathbb{E}[L]_l > \mathbb{E}[L]$ holds asymptotically as $n \to \infty$, provided that $\gamma n^2 \leq C$, i.e., $\gamma = O(1/n^2)$. See proof in Appendix A.*

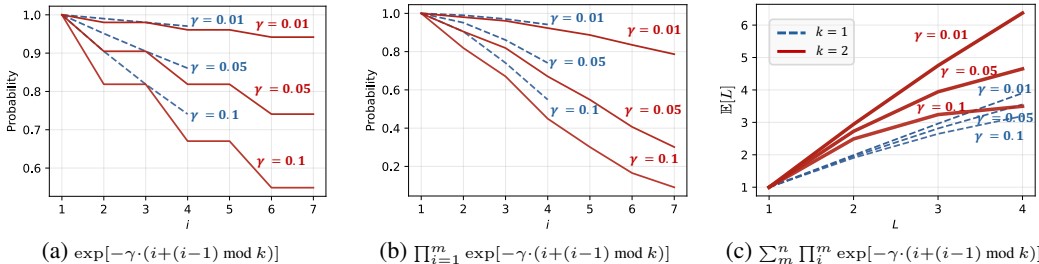

(a) $\exp[-\gamma \cdot (i+(i-1) \bmod k)]$  (b) $\prod_{i=1}^{m} \exp[-\gamma \cdot (i+(i-1) \bmod k)]$  (c) $\sum_{m}^{n} \prod_{i}^{m} \exp[-\gamma \cdot (i+(i-1) \bmod k)]$

Figure 5: Different curves of expected marginal probability (a), joint probability (b), and accepted length (c), with $k \in \{1, 2\}$. The leap strategy extends the range of prediction at position $i$, and achieves a higher length of expectation.

**Remark.** L-MTP introduces a longer prediction range $(k(n-1)+1)$ to compensate for the loss of confidence on leaping positions. Theorem 3 reveals the relation between attenuation and the number of prediction heads. Less attenuation indicates higher speed-up of L-MTP compared to the vanilla strategy. In practice, $n$ would not be too large. Less attenuation ($\gamma$) with fewer overheads of heads ($n$) leads to further higher speed up. We illustrate the simulated curves for a better understanding of the superiority of L-MTP (see Figure 5).

**Illustration of analyses.** To intuitively demonstrate the effectiveness of our method, we provide the illustration of the above theoretical analyses as shown in Figure 5. We simulate the probabilities and expectations of length and observe that L-MTP ($k > 1$) outperforms MTP ($k = 1$) given different attenuation cases. Less attenuation leads to higher speedup of L-MTP.

## 5 Experiments

In this section, we conduct experiments to address the following research question:

- **RQ1:** How does L-MTP perform on different LLM tasks compared to other prediction paradigms? Can it benefit the training of LLMs, thus boosting model performance?

- **RQ2:** Can L-MTP bring further inference acceleration by the decoding strategy of looking backward? Is L-MTP's decoding strategy extendable to further models?

- **RQ3:** What is the prediction accuracy of each output head? Does it satisfy our theoretical analyses in Section 4?

- **RQ4:** What is the potential of L-MTP? Does it suggest further findings on different scales of models and data?

### 5.1 Experimental Setup

**Base LLMs.** The experiments utilize the following base large language models: *Qwen 2.5* (3B and 7B parameters), *Llama 3.2* (3B parameters), *Llama 3.1* (8B parameters), and *Gemma 3* (4B and 12B parameters). These models are selected to represent a diverse range of architectures and parameter scales for comprehensive evaluation. We elaborate on more details in Appendix B.2.

**Baselines.** For efficacy comparison, we evaluate two prediction paradigms: next token prediction (NTP) and multi-token prediction (MTP). These paradigms assess the models' ability to generate accurate and contextually relevant outputs under different prediction strategies. For efficiency, we use NTP with autoregressive decoding as the basis. We compare L-MTP with MTP, which leverages self-speculative decoding, and a trivial forward decoding way (see the implementation in Appendix C.1), to analyze the inference efficiency.

**Datasets.** We curate the training dataset from *Math* [53], *Evol-Instruct-Code* [54, 55], and *Alpaca-GPT4* [56]. In the first stage, we use the full data for self-distillation. For the second stage, we randomly select 10,000 examples with a ratio of 4:4:2, corresponding to math, code, and general data, respectively. To benchmark the methods, we select *Math500* [57] (4-shot) and *GSM8K* [58] (4-shot) for math evaluation, *MBPP*, *MBPP$^+$* [59, 60], *HumanEval*, and *HumanEval$^+$* [61, 60] for

Table 1: Performance comparison with different prediction paradigms across diverse tasks and benchmarks. In each case, the best average result (Avg.) by comparing among NTP, MTP, and L-MTP is demonstrated in bold.

| | | Math500 | GSM8K | MBPP | MBPP$^+$ | HumanEval | HumanEval$^+$ | MMLU | IFEval | Avg. |
|---|---|---|---|---|---|---|---|---|---|---|
| *Llama3.2-3B* | Base | 2.20 | 1.06 | 50.00 | 40.48 | 27.44 | 24.39 | 54.23 | 18.23 | 27.25 |
| | NTP | 3.00 | 3.71 | 47.09 | 36.24 | 21.34 | 17.68 | 54.34 | 20.74 | 25.52 |
| | MTP | 3.40 | 3.87 | 46.83 | 36.51 | 21.95 | 18.29 | 54.22 | 18.59 | 25.46 |
| | L-MTP | 4.80 | 5.91 | 46.56 | 36.51 | 24.39 | 20.73 | 54.17 | 20.38 | **26.68** |
| *Llama3.1-8B* | Base | 4.20 | 9.86 | 61.38 | 51.32 | 39.02 | 31.71 | 63.26 | 18.23 | 34.87 |
| | NTP | 5.60 | 11.30 | 61.38 | 51.06 | 42.68 | 35.37 | 63.64 | 20.14 | 36.40 |
| | MTP | 6.40 | 10.08 | 60.32 | 49.74 | 41.46 | 35.98 | 63.52 | 19.42 | 35.87 |
| | L-MTP | 6.40 | 10.92 | 61.38 | 50.53 | 42.68 | 36.59 | 63.70 | 22.18 | **36.80** |
| *Qwen2.5-3B* | Base | 35.40 | 53.75 | 62.70 | 53.97 | 68.29 | 61.59 | 65.13 | 32.73 | 54.20 |
| | NTP | 25.40 | 49.13 | 66.93 | 57.94 | 67.68 | 60.98 | 65.17 | 34.17 | 53.43 |
| | MTP | 25.40 | 45.79 | 67.72 | 57.67 | 65.85 | 59.15 | 65.21 | 35.49 | 52.79 |
| | L-MTP | 28.20 | 46.25 | 67.99 | 59.26 | 67.68 | 60.37 | 65.23 | 35.01 | **53.75** |
| *Qwen2.5-7B* | Base | 63.00 | 56.79 | 75.93 | 65.34 | 78.05 | 71.34 | 71.93 | 42.69 | 65.63 |
| | NTP | 49.40 | 52.99 | 78.31 | 67.46 | 78.05 | 69.51 | 71.78 | 43.41 | 63.86 |
| | MTP | 49.00 | 52.62 | 78.04 | 67.99 | 76.22 | 69.51 | 71.85 | 41.49 | 63.34 |
| | L-MTP | 46.00 | 56.03 | 78.04 | 67.72 | 77.44 | 71.95 | 71.98 | 44.12 | **64.16** |
| *Gemma3-4B* | Base | 0.00 | 0.00 | 60.58 | 51.59 | 33.54 | 28.05 | 38.21 | 26.50 | 29.81 |
| | NTP | 6.20 | 4.70 | 58.20 | 51.06 | 46.34 | 39.02 | 58.29 | 35.49 | **37.41** |
| | MTP | 6.00 | 4.32 | 58.47 | 50.53 | 43.29 | 37.20 | 58.25 | 34.65 | 36.59 |
| | L-MTP | 7.60 | 4.25 | 57.67 | 49.47 | 45.73 | 38.41 | 58.33 | 34.65 | 37.01 |
| *Gemma3-12B* | Base | 0.00 | 9.78 | 73.28 | 59.52 | 45.73 | 36.59 | 23.79 | 29.38 | 34.76 |
| | NTP | 10.00 | 13.42 | 71.16 | 59.79 | 63.41 | 56.10 | 71.69 | 29.38 | 46.87 |
| | MTP | 9.20 | 5.61 | 70.11 | 58.47 | 61.59 | 54.27 | 71.67 | 30.46 | 45.17 |
| | L-MTP | 17.20 | 26.38 | 70.11 | 60.05 | 62.20 | 55.49 | 72.10 | 33.09 | **49.58** |

code evaluation, and *MMLU* [62] and *IFEval* [63] for general evaluation. We detail the statistics and utilization of these datasets in Appendix B.3, Appendix B.4, and Appendix B.6.

**Evaluation metrics.** For performance comparison, we utilize *accuracy* for both math and general tasks and *pass@1* for code tasks. For efficiency analysis, we employ the *speedup ratio* as the metric, which is calculated by the relative generated tokens per second compared to the original. Higher values indicate better performance.

**Implementation details.** To adapt L-MTP for NTP-based methods, we employ a two-stage training procedure. At the head warming up stage, we freeze the LLM backbone while training the heads with a learning rate of $1\times10^{-3}$ for 5 epochs. We utilize the cosine scheduler and set the warmup ratio as 0.1. At the next stage, we utilize LoRA [64] with rank being 32 and alpha being 16 to tune the full model. Here we only train the model for 3 epochs with the learning rate being $1\times10^{-5}$. We set $k = 2$ and $n = 4$ by default. This training setting is also employed for MTP implementation to ensure fairness. We also provide the pseudo-code of L-MTP in Appendix B.1. All the experiments are conducted on $2\times$ NVIDIA H100-80G GPUs.

## 5.2 Results and Discussions

**Overall performance (RQ1).** To answer RQ1, we compare L-MTP with MTP and NTP across diverse datasets and involve a range of base models as backbones, as shown in Table 1. Through the comparison, we can observe the improvement brought by L-MTP for different scales and series of models, especially on math tasks for the Llama and Gemma series, and code tasks for the Qwen series. Furthermore, we find all models gain improvement on general tasks, exemplified by IFEval. Notably, L-MTP achieves better performance for most tasks compared to MTP. Intriguingly, we observe that in some cases, even NTP also brings worse results. Although L-MTP can compensate for the margin, the deterioration still cannot be mitigated. Carefully choosing higher-quality data would

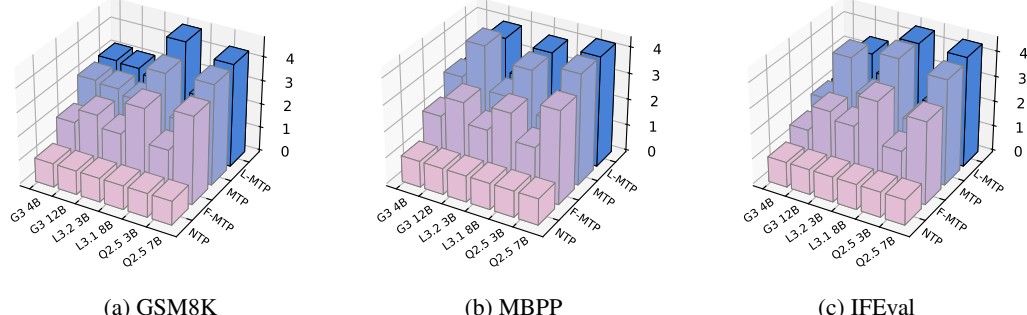

(a) GSM8K  (b) MBPP  (c) IFEval

Figure 6: Speedup with self-speculative decoding for different series of LLMs ("G"↔Gemma, "L"↔Llama, and "Q"↔Qwen). The Z-axis represents the speedup ratio.

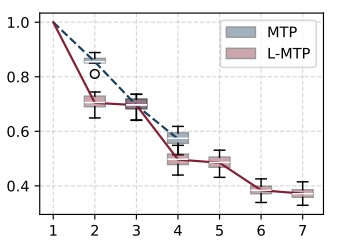
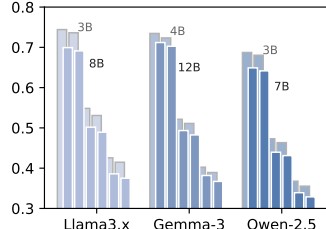
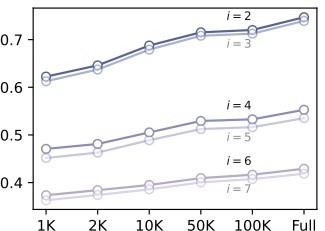

Figure 7: The prediction accuracy at different positions estimated on the alpaca-eval split.

Figure 8: The prediction accuracy for different models. Myopia worsens as scale increases.

Figure 9: The prediction accuracy improves as training data increases (from 1K to the full).

be beneficial. However, in this paper, we do not focus on how to select data, but on investigating the effect of L-MTP compared to MTP. Such a phenomenon also motivates us to explore more in-depth analyses and discussions.

**Inference acceleration (RQ2).** L-MTP implements the decoding by looking backward to achieve inference acceleration without any architecture modifications or complex operations. We provide the inference speedup comparison as shown in Figure 6. We also implement a trivial solution for leaping prediction by looking forward, denoted as F-MTP. We provide more implementation details in Appendix C.1. Compared to MTP, L-MTP achieves comparable yet sometimes higher speedup, especially on GSM8K. L-MTP predicts the farther position, while leaving the blank filled by the previous prediction, thus achieving faster inference.

We also explore the potential by extending L-MTP decoding to existing models, like Medusa [48], which is specialized for improving the acceptance rate for multiple heads. We equip these models with L-MTP decoding and showcase the results in Table 2. Directly changing the decoding strategy to the leaping paradigm brings up to 1.3× speed up (22% relative boosting). These results demonstrate the potential of L-MTP, especially for models with higher acceptance rates.

Table 2: The speed up ratio comparison when extending L-MTP to Medusa on different scales of models.

|  |  | GSM8K | MBPP |
|---|---|---|---|
| *Viccuna 7B* | MTP | 1.83× | 1.97× |
|  | L-MTP | **2.32×** | **2.01×** |
| *Viccuna 13B* | MTP | 2.24× | 1.98× |
|  | L-MTP | **2.43×** | **2.02×** |

**The expected distribution at each position (RQ3).** We calculate the prediction accuracies at each position to verify our theoretical analyses in Section 4. We plot the different accuracies for different models (box), and show the average ones at each position (line) for both MTP and L-MTP in Figure 7. These practical results manifest the property of Attenuation (*cf.*, Definition 1) and Consistency (*cf.*, Assumption 2), and resemble our simulated illustration, particularly providing a strong support for our theoretical analyses.

**Comparison to MTP with $n = 7$.** Observed from Table 3, we can see that directly increasing the horizon of MTP does not improve the performance overall. But we can still observe some improvement on HumanEval. The interesting thing is that when we decrease the number of heads to 3, L-MTP ($k = 3, n = 3$) can achieve a better performance than MTP ($n = 7$). The theory on the prediction analysis at different positions can answer this (Section 4). Distant tokens would lead to

Table 3: Performance comparison to MTP *w.r.t.* different prediction horizons across diverse tasks and benchmarks. MTP $_{n=7}$, L-MTP $_{k=2,n=4}$, and L-MTP $_{k=3,n=3}$ can predict the next 7 tokens for one-time decoding.

|  | Math500 | GSM8K | MBPP | MBPP+ | HumanEval | HumanEval+ | MMLU | IFEval | Avg. |
|---|---|---|---|---|---|---|---|---|---|
| MTP$_{n=4}$ | 25.40 | 45.79 | 67.72 | 57.67 | 65.85 | 59.15 | 65.21 | 35.49 | 52.79 |
| MTP$_{n=7}$ | 24.40 | 43.29 | 63.49 | 55.29 | 68.29 | 61.59 | 65.11 | 33.09 | 51.82 |
| L-MTP$_{k=2,n=4}$ | 28.20 | 46.25 | 67.99 | 59.26 | 67.68 | 60.37 | 65.23 | 35.01 | **53.75** |
| L-MTP$_{k=3,n=3}$ | 28.00 | 51.86 | 60.05 | 52.65 | 66.46 | 62.20 | 65.06 | 32.49 | 52.35 |

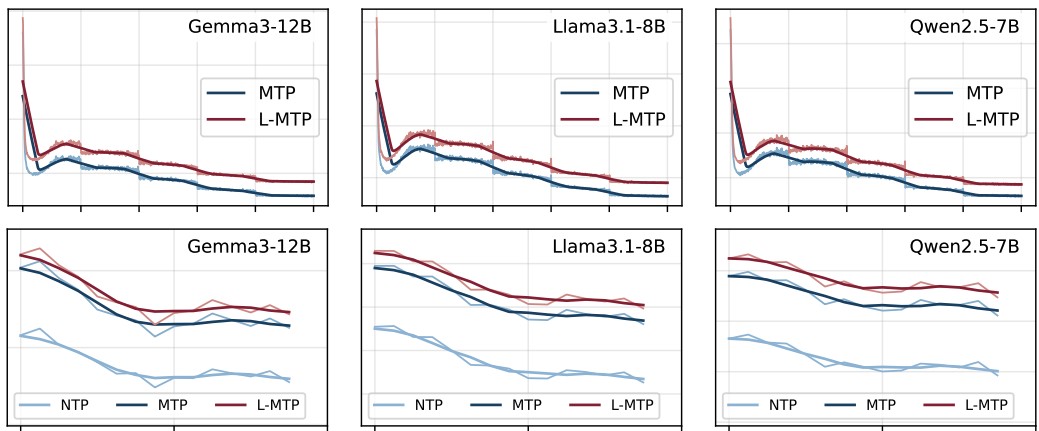

Figure 10: The illustration of the warm-up procedure (*top*) and full model fine-tuning (*bottom*) of adapting multiple output heads to LLMs. The curves showcase the loss changes along with the training for NTP, MTP, and L-MTP.

noise, while our leap can catch future tokens, but also leap some. An accumulated noise would be smaller than MTP. A smaller number of heads ($n = 3$) while with the leaping strategy can achieve a comparable performance (52.79, MTP with $n = 4$) or outperform MTP (51.82, $n = 7$).

**Training loss curves.** We showcase the training loss trends across different model series, *i.e.*, Gemma3-12B, Llama3.1-8B, and Qwen2.5-7B, for the two-stage training. As shown in Figure 10, we observe the losses steadily decreasing and convergence to stability, even if L-MTP predicts further positions with the same overhead. To endow the pre-trained model with multi-head prediction capability, we also observe a turning point, indicating the myopia of models. For other smaller models, we also present the training loss trends in Figure 12 for head warm-up and Figure 13 for full model tuning (*cf.* Appendix D).

**Potential analysis (RQ4).** We emphasize the potential of L-MTP by investigating the myopia of LLMs and the effect of data amount. *(1) Myopic generation.* We demonstrate the prediction accuracy across different scales of models, as shown in Figure 8. The accuracy drops consistently when changing the small model to the larger one for all series of LLMs, indicating the inherent myopia imposed by NTP pre-training. We also provide the loss curves during training in Appendix D, which show an inflection when warming up the heads. Recent work [39] suggests training a model from scratch with the MTP objective. This is also promising for L-MTP to inherently benefit the model with a broader range of predictions and faster inference. *(2) Data scales.* We illustrate the increasing prediction accuracy when adding more data in Figure 9 at the head warming up stage. Large-scale data introduces more diversity to help additional heads adapt the LLM backbone. However, we also observe that the increase is not linear. To obtain higher accuracy, equipping L-MTP with more sophisticated techniques for training or model architecture [65, 52] will also be promising.

## 6   Related Work

**Multi-token prediction.** Previous studies demonstrate that multi-token prediction (MTP) encourages pre-planning capabilities in large language models (LLMs). Qi *et al.* [38] pioneer $n$-step-ahead prediction to optimize language models, mitigating overfitting to strong local dependencies. Gloeckle *et al.* [39] pretrain LLMs with additional prediction heads to achieve significant perfor-

mance improvements, particularly on code-related tasks. Industrial deployments have also adopted MTP to improve both training efficiency and pre-planning [40, 41]. MTP has sparked growing interest in exploring its potential, including adapting next-token prediction (NTP) models for MTP [66] and applying it to domains such as speech [67]. Furthermore, recent work investigates MTP's potential for inference acceleration by incorporating additional prediction heads, as exemplified by Medusa [48]. We discuss LLM inference acceleration further in the next subsection.

In addition, recent research on MTP shows significant promise, with the support of LLMs inherently maintaining certain pre-planning [68]. These methods typically assume prediction within an adjacent context for LLMs, predicting the next $n$ tokens simultaneously at each time step. We go beyond its prediction pattern and introduce leaps between prediction tokens, therefore extending a broader training signal.

**LLM inference acceleration.** There is a bunch of work focusing on accelerating LLMs, especially on their inference procedure [69, 70, 71, 72]. The remarkable techniques involve quantization [73, 74], pruning [75, 76], knowledge distillation [77, 56], compact architecture design [78, 79, 80], and dynamic network [81, 82]. The production deployment also advances to improve the inference efficiency, like memory management [83, 84] and parallelism [85, 86]. In this paper, we focus on the inference acceleration benefited by LLM decoding.

Prior works accelerate inference on greedy decoding [87, 88], while recent speculative decoding extends it with provably losslessness [49, 89, 90]. Speculative decoding follows the principle of *draft-then-verify*, where a draft model (smaller) efficiently generate multiple tokens for the parallel verification via the target model (larger). The drafting procedure can employ an independent model [50, 91] or enhance the own model [92], like adding additional FFN heads [87, 48, 52]. During the verification, the vanilla sampling only process tokens in a single draft sequence [49, 93], while recent methods utilize the token tree to verify multiple draft sequences in parallel [50, 51, 48, 52], further improve the token acceptance rate. In L-MTP, we apply our looking backward decoding by utilizing the additional LLM heads for self-speculative decoding, paired with the tree-based verification.

# 7    Conclusion

In this paper, we propose leap multi-token prediction as an improvement over vanilla multi-token prediction in the training and inference of large language models for generative or reasoning tasks. Both theoretical insights and empirical evidence are offered to justify the superiority of the proposed method, where both model performance and inference speed can be enhanced simultaneously in a series of scenarios. In future work, we would like to better understand how to adaptively choose $n$ and $k$ in leap multi-token prediction losses. One possibility is to determine their values based on the local uncertainty or entropy of the predicted tokens, which allows the model to leap more aggressively in low-entropy regions while maintaining finer granularity in more ambiguous contexts. Also, reinforcement fine-tuning has emerged as a promising paradigm for training large language models. Incorporating our method into this training framework opens up exciting opportunities and is worth further exploration.

# Acknowledgements

This research/project is supported by the National Research Foundation, Singapore under its National Large Language Models Funding Initiative (AISG Award No: AISG-NMLP-2024-002). Any opinions, findings and conclusions or recommendations expressed in this material are those of the author(s) and do not reflect the views of National Research Foundation, Singapore.

Xiaobo Xia is supported by MoE Key Laboratory of Brain-inspired Intelligent Perception and Cognition, University of Science and Technology of China (Grant No. 2421002). Xiu Su is supported by National Natural Science Foundation of China (No. 62406347). Shuo Yang is supported by the NSFC Young Scientists Fund (No. 62506096).

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

# Appendix

# A  Detailed Proof of Theorem 3

For LLMs with $n$ output heads that predict multiple tokens at once, the expectation of the accepted length can be represented as:

$$\mathbb{E}[L] = \sum_{m=1}^{n} \left( \prod_{i=1}^{m} \mathbb{E}_{\mathrm{x}_{\leq t} \sim \mathcal{D}}[p(\mathrm{x}_{t+i}|\mathrm{x}_{\leq t})] \right) \quad \text{(Vanilla)}. \tag{8}$$

In this case, we only utilize the last hidden state, resulting in $n$ tokens $\mathrm{x}_{t+n:t+1}$. Without changing the original probabilities, we propose a decoding strategy by looking backward that yields reorganized expectations:

$$\mathbb{E}[L]_b = \sum_{m=1}^{k(n-1)+1} \left( \prod_{i=1}^{m} \mathbb{E}_{\mathrm{x}_{\leq t} \sim \mathcal{D}}[p(\mathrm{x}_{t+i}|\mathrm{x}_{\leq t-(i-1) \bmod k})] \right) \quad \text{(L-MTP)}. \tag{9}$$

**Proof of Theorem 3.**

*Proof.*

$$\mathbb{E}[L]_b = \sum_{m=1}^{k(n-1)+1} \prod_{i=1}^{m} \mathbb{E}_{\mathrm{x}_{\leq t} \sim \mathcal{D}}[p(\mathrm{x}_{t+i}|\mathrm{x}_{\leq t-(i-1) \bmod k})] \tag{10}$$

$$= \sum_{m=1}^{k(n-1)+1} \prod_{i=1}^{m} f(i + (i-1) \bmod k) \tag{11}$$

$$= \sum_{m=1}^{n} \prod_{i=1}^{m} f(i + (i-1) \bmod k) + \sum_{m=n+1}^{k(n-1)+1} \prod_{i=1}^{m} f(i + (i-1) \bmod k) \tag{12}$$

$$\underbrace{\mathbb{E}[L]_b - \mathbb{E}[L]}_{\boldsymbol{\Delta}_b} = \sum_{m=1}^{k(n-1)+1} \prod_{i=1}^{m} p(\mathrm{x}_{t+i}|\mathrm{x}_{\leq t-(i-1) \bmod k}) - \sum_{m=1}^{n} \prod_{i=1}^{m} p(\mathrm{x}_{t+i}|\mathrm{x}_{\leq t}) \tag{13}$$

$$= \underbrace{\sum_{m=1}^{n} \left[ \prod_{i=1}^{m} f(i + (i-1 \bmod k)) - \prod_{i=1}^{m} f(i) \right]}_{\boldsymbol{\Delta}_b^1} + \underbrace{\sum_{m=n+1}^{k(n-1)+1} \prod_{i=1}^{m} f(i + (i-1 \bmod k))}_{\boldsymbol{\Delta}_b^2}. \tag{14}$$

Here $\boldsymbol{\Delta}_b$ is the difference between two expectations, expressed as sums involving products of probabilities, with a function $f(i)$ that decreases as $i$ increases. Besides, $k \leq n$, which means the stride $k$ is at most the number of heads $n$. That $f(i)$ is a monotonically decreasing function, meaning $f(i) \geq f(j)$ for $i < j$. Formally,

$$f(i + (i-1 \bmod k)) \leq f(i), \text{with equality only when } i \bmod k \equiv 1. \tag{15}$$

Therefore,

$$\boldsymbol{\Delta}_b = \boldsymbol{\Delta}_b^1 + \boldsymbol{\Delta}_b^2, \quad \boldsymbol{\Delta}_b^1 \leq 0, \quad \boldsymbol{\Delta}_b^2 > 0. \tag{16}$$

For $\boldsymbol{\Delta}_b > 0$, the positive $\boldsymbol{\Delta}_b^2$ must outweigh the negative $\boldsymbol{\Delta}_b^1$. The decay rate, controlled by $\gamma$, and the number of terms, controlled by $n$, will determine this balance.

To resolve the conditions for $\boldsymbol{\Delta}_b > 0$, we assume $f(i) \sim \Theta(1/\mathrm{poly}(i))$, *i.e.*, $f(i) = 1/\exp[\gamma \cdot (i-1)]$, where $\gamma > 0$ is the attenuation coefficient.

$$\boldsymbol{\Delta}_b^1 = \sum_{m=1}^{n} \left[ \prod_{i=1}^{m} \frac{1}{\exp[\gamma(i + (i-1 \bmod k))]} - \prod_{i=1}^{m} \frac{1}{\exp[\gamma \cdot i]} \right]. \tag{17}$$

$$\boldsymbol{\Delta}_b^2 = \sum_{m=n+1}^{kn-1} \prod_{i=1}^{m} \frac{1}{\exp[\gamma(i + (i-1 \bmod k))]}. \tag{18}$$

$$\boldsymbol{\Delta}_b = \sum_{m=1}^{n} \left[ \prod_{i=1}^{m} \frac{1}{\exp[\gamma(i + (i-1 \bmod k))]} - \prod_{i=1}^{m} \frac{1}{\exp[\gamma \cdot i]} \right] \tag{19}$$

$$+ \sum_{m=n+1}^{kn-1} \prod_{i=1}^{m} \frac{1}{\exp[\gamma(i + (i-1 \bmod k))]} \tag{20}$$

$$= \sum_{m=1}^{n} \left[ \exp[-\gamma \sum_{i=1}^{m} (i + (i-1 \bmod k))] - \exp[-\gamma \sum_{i=1}^{m} i] \right] \tag{21}$$

$$+ \sum_{m=n+1}^{kn-1} \exp[-\gamma \sum_{i=1}^{m} (i + (i-1 \bmod k))]. \tag{22}$$

We resolve it in the case for $k = 2$. Specifically,

$$\boldsymbol{\Delta}_b = \sum_{m=1}^{n} \left( \exp[-\gamma \sum_{i=1}^{m} (i + (i-1 \bmod 2))] - \exp[-\gamma \sum_{i=1}^{m} i] \right) \tag{23}$$

$$+ \sum_{m=n+1}^{2n-1} \exp[-\gamma \sum_{i=1}^{m} (i + (i-1 \bmod 2))] \tag{24}$$

$$= \sum_{m=1}^{n} \left( \exp[-\gamma \frac{m(m+1)}{2} + \left\lfloor \frac{m}{2} \right\rfloor] - \exp[-\gamma \frac{m(m+1)}{2}] \right) \tag{25}$$

$$+ \sum_{m=n+1}^{2n-1} \exp[-\gamma (\frac{m(m+1)}{2} + \left\lfloor \frac{m}{2} \right\rfloor)] \tag{26}$$

$$= \sum_{m=1}^{n} \exp[-\gamma \frac{m(m+1)}{2}] \left( \exp[-\gamma \left\lfloor \frac{m}{2} \right\rfloor] - 1 \right) + \sum_{m=n+1}^{2n-1} \exp[-\gamma (\frac{m(m+1)}{2} + \left\lfloor \frac{m}{2} \right\rfloor)] \tag{27}$$

Consider the upper bound of $|\boldsymbol{\Delta}_b^1|$:

$$|\boldsymbol{\Delta}_b^1| = \sum_{m=1}^{n} \exp[-\gamma \frac{m(m+1)}{2}] \left( 1 - \exp[-\gamma \left\lfloor \frac{m}{2} \right\rfloor] \right) \tag{28}$$

$$\leq \sum_{m=1}^{n} \gamma \left\lfloor \frac{m}{2} \right\rfloor \exp[-\gamma \frac{m(m+1)}{2}] \quad (1 - \exp(-x) \leq x, \forall x \geq 0) \tag{29}$$

$$\leq \frac{\gamma}{2} \sum_{m=1}^{n} m \exp[-\gamma \frac{m(m+1)}{2}] \quad (\left\lfloor \frac{m}{2} \right\rfloor \leq \frac{m}{2}) \tag{30}$$

$$\leq \frac{\gamma}{2} \int_{0}^{n+1} x \exp[-\gamma \frac{x^2}{2}] dx \tag{31}$$

$$= \frac{\gamma}{2} \cdot \frac{1}{\gamma} \int_{0}^{\sqrt{\gamma}(n+1)} y \exp[-\frac{y^2}{2}] dy \quad (\text{let } y = \sqrt{\gamma} x) \tag{32}$$

$$= \frac{\gamma}{2} \cdot \frac{1}{\gamma} \left( 1 - \exp[-\gamma \frac{(n+1)^2}{2}] \right) \quad (\int_{0}^{a} y e^{-y^2/2} dy = 1 - e^{-a^2/2}) \tag{33}$$

$$= \frac{1}{2} \left( 1 - \exp[-\gamma \frac{(n+1)^2}{2}] \right). \tag{34}$$

Afterward, consider the lower bound of $|\mathbf{\Delta}_b^2|$:

$$\mathbf{\Delta}_b^2 = \sum_{m=n+1}^{2n-1} \exp[-\gamma(\frac{m(m+1)}{2} + \left\lfloor \frac{m}{2} \right\rfloor)] \tag{35}$$

$$\geq \sum_{m=n+1}^{2n-1} \exp\left[-\gamma \frac{m^2}{2}\right] \tag{36}$$

$$\geq \int_{n+1}^{2n} \exp[-\gamma \frac{x^2}{2}] dx \tag{37}$$

$$\geq \frac{1}{\sqrt{\gamma}} \int_{\sqrt{\gamma}(n+1)}^{\sqrt{\gamma} \cdot 2n} \exp[-\frac{y^2}{2}] dy \quad (y = \sqrt{\gamma} x) \tag{38}$$

$$\geq \frac{1}{\sqrt{\gamma}} \left( \frac{\exp[-2\gamma n^2]}{2\gamma n} - \frac{\exp[-\gamma(n+1)^2]}{\gamma(n+1)} \right). \tag{39}$$

$$(\int_a^b \exp[-y^2/2] dy \geq \frac{\exp[-b^2/2]}{b} - \frac{\exp[-a^2/2]}{a}, \quad \text{for } b > a > 0)$$

$$\gtrsim \frac{1}{\sqrt{\gamma}} \cdot \frac{\exp[-2\gamma n^2]}{2n}. \tag{40}$$

Substitute the bounds:

$$\mathbf{\Delta}_b^2 > |\mathbf{\Delta}_b^1| \Rightarrow \frac{1}{\sqrt{\gamma}} \cdot \frac{\exp[-2\gamma n^2]}{2n} > \frac{1}{2}(1 - \exp[-\gamma \cdot \frac{(n+1)^2}{2}]).$$

Introducing $\beta = \gamma n^2$, this becomes:

$$\frac{\exp[-2\beta]}{n\sqrt{\beta}} > \frac{\beta}{2} \Rightarrow \exp[-2\beta] > \frac{n\beta^{3/2}}{2} \Rightarrow \frac{\exp[-2\beta]}{\sqrt{\beta}} > \beta \Rightarrow \exp[-2\beta] > \beta^{3/2} \Rightarrow 1 > \beta^{3/2} e^{2\beta}.$$

For the inequality to hold, it suffices that $\beta = O(1)$, which implies $\gamma = O\left(1/n^2\right)$. $\qquad \square$

# B    Implementation Details

## B.1    Pseudo-Code for L-MTP

We provide the training and inference pseudo-code for L-MTP. For training, we only need to add a leap control $k$ to reassign the prediction positions to modify MTP to L-MTP.

**Training of L-MTP**

```python
# multi-head forward computing >>>
for i in range(self.n_head):
    logits.append(self.heads[i](hidden_states))
# multi-head forward computing <<<
# ....
# Leap Multi-token Prediction >>>
# if k == 1: L-MTP = MTP
for i, logits_ in enumerate(logits):
    h_logits = logits_[:, : -(k*(i+1))].contiguous()
    h_labels = labels[..., k*(i+1) :].contiguous()
    loss_i = self.loss_fct(logits=h_logits, labels=h_labels,
        vocab_size=self.config.vocab_size)
    loss += loss_i
# Leap Multi-token Prediction <<<
# ...
```

For inference, we add a cache to store the previous hidden state for our decoding. The decoding involves both previous and current hidden states to yield $k(n-1)$ additional tokens.

**Inference of L-MTP**

```python
# multi-head forward computing >>>
for i in range(self.n_head):
    logits.append(self.heads[i](hidden_states))
# multi-head forward computing <<<
# ....

# get the previous one
if self.model.past_hidden_states is None:
    self.model.past_hidden_states =
        hidden_states[:, -2].unsqueeze(1)

past_hidden_states = torch.stack([self.model.past_hidden_states,
    hidden_states[:, -1].unsqueeze(1)], dim=0)
logits_ = self.heads(past_hidden_states)
lmtp_logits = logits_.flatten(start_dim=0, end_dim=1)
# update the past hidden states
self.model.past_hidden_states = past_hidden_states[-1]
# ...
```

## B.2 Base LLMs

The experiments leverage a diverse set of base large language models (LLMs) to ensure a comprehensive evaluation across varying architectures and parameter scales. Below, we introduce the selected models: **Qwen 2.5** (3B and 7B) [31] developed by Alibaba Cloud, **Llama 3.2** (3B), **Llama 3.1** (8B) [2] developed by Meta AI, and **Gemma 3** (4B and 12B) [94] developed by Google.

## B.3 Training Datasets

**Math**[4] [53]: This dataset comprises a curated collection of mathematical problems and solutions, spanning topics such as algebra, calculus, geometry, and discrete mathematics. It is designed to enhance the reasoning and problem-solving capabilities of large language models, particularly in numerical and symbolic computation tasks. We utilize the training dataset with 7.5K problems.

---

[4]https://github.com/hendrycks/math

**Evol-Instruct-Code**[5] [54, 55]: This dataset is an evolved version of instruction-based code generation data, built upon iterative refinement and augmentation techniques. It contains a wide range of programming tasks, solutions, and explanatory instructions across multiple languages (*e.g.*, Python, Java, C++). This dataset is curated following the code generation instruction process described in the WizardCoder [54]. It is based on the Code Alpaca 20k dataset [55] and evolves each instruction through a randomly chosen evolution prompt, see more details in its public repository [6]. As a result, this dataset is constructed with 80K examples with the merging of the seed dataset and three evolutions.

**Alpaca-GPT4**[7] [56]: The dataset is a collection of English instruction-following data generated by GPT-4 using Alpaca prompts, specifically designed for fine-tuning LLMs. This dataset is a derivative of the original Alpaca dataset and leverages the same prompts but uses GPT-4 to generate the completions, resulting in higher quality and more detailed responses. The original Alpaca dataset used text-davinci-003 to complete the prompts. In contrast, Alpaca-GPT4 uses GPT-4, resulting in more detailed and higher-quality responses. The dataset consists of 52K unique instruction-following examples.

## B.4 Evaluation Benchmarks

**MATH500** [57] is a subset of the MATH dataset, comprising 500 challenging mathematical problems designed to test advanced mathematical reasoning and problem-solving skills. It includes problems from various domains such as algebra, calculus, geometry, and number theory, primarily at high school and early undergraduate levels.

**GSM8K** [58] is a dataset of grade-school-level math word problems. It focuses on elementary arithmetic, basic algebra, and logical reasoning, which requires models to understand natural language descriptions and perform multi-step calculations. We utilize its test split, which contains 1,319 examples in total.

**MBPP & MBPP**$^+$ [59, 60] is a dataset of 974 Python programming problems designed to evaluate code generation and problem-solving abilities. Tasks range from simple functions to moderately complex algorithms, requiring correct implementation in Python. MBPP$^+$ adds more unique test-cases ($30\times$) from the original MBPP [60].

**HumanEval & HumanEval**$^+$ [61, 60] is a dataset of 164 hand-crafted Python programming problems, focusing on evaluating the functional correctness of code generation. Each problem includes a function signature, description, and test cases to verify the solution. HumanEval$^+$ adds more unique test-cases ($80\times$) and fixes incorrect ground-truth solutions in HumanEval [60].

**MMLU** [62] is a comprehensive benchmark consisting of 57 tasks covering topics from STEM (science, technology, engineering, and mathematics), humanities, social sciences, and professional fields. The tasks are multiple-choice questions at high school, college, and professional levels, designed to evaluate the model's broad knowledge and reasoning capabilities. Performance is measured by accuracy across all tasks.

**IFEval** [63] is a dataset designed to assess a model's ability to follow explicit instructions in natural language. It includes a variety of tasks where models must generate responses that adhere strictly to given guidelines, such as formatting, content constraints, or specific reasoning steps.

## B.5 Head Architecture

We describe the head architecture of Medusa [48], which is also adopted in our implementation. Specifically, given the hidden state $\mathbf{z}$ at the last layer of LLMs, the head will first transform it via $\mathbf{z}' = \mathbf{z} + \text{SiLU}(\mathbf{W}\mathbf{z} + \mathbf{b})$, where $\mathbf{W} \in \mathbb{R}^{d\times d}$, $\mathbf{b} \in \mathbb{R}^{d\times 1}$, $d$ is the dimension of hidden state and SiLU is the a Sigmoid Linear Unit (SiLU) function, denoted as $\text{SiLU}(\mathbf{x}) = \mathbf{x} \cdot \sigma(\mathbf{x})$. After that,

---

[5]https://huggingface.co/datasets/ise-uiuc/Magicoder-Evol-Instruct-110K
[6]https://github.com/nickrosh/evol-teacher
[7]https://github.com/Instruction-Tuning-with-GPT-4/GPT-4-LLM

the transformed hidden state is mapped to the logits, with the output dimensions being the size of the vocabulary. Such a process can be formulated as $\mathbf{W}_{\text{head}}\mathbf{z}'$. Notably, $\mathbf{W}_{\text{head}}$ is initialized with the weight of the original head of the backbone LLM, and $\mathbf{W}$ is initialized with zeros.

### B.6 Data Curation

**Self-distillation** [95]. At the head warm-up stage, the main goal is to align additional heads with the original head to improve acceptance rates, as also suggested in [48]. Therefore, we employ the self-distillation strategy for different backbone LLMs. In this case, we use vLLM [8] for efficiently generating the responses for every data point. The generated dataset will be stored and then serve as the training data for warm-up training.

**Downsampling** [96, 70]. At the continued training stage, we downsample the dataset randomly, where we take 4,000 examples for both code and math datasets and 2,000 examples for the general dataset. Therefore, we prepare 10K examples for continuing to train the model. We keep the curated dataset fixed for a fair comparison in our experiments.

## C  Decoding Strategy

### C.1  Forward Decoding

We also provide another trivial alternative by looking *forward*, denoted F-MTP. For instance, F-MTP predicts tokens $\{x_{t+k(i-1)+2}\}_{i\in[n]}$ are predicted given $x_{\leq t+1}$. In this case, we have:

$$\left\{ p(x_{t+i}|x_{<t+(i-1)\bmod k})|i \in \{0, 1, \ldots, kn\} \right\},\tag{41}$$

where the token sequence is sampled by looking forward $(+)$ $k-1$ steps. Forward decoding prioritizes early tokens, resulting in tokens $\{x_{t+k}\}$ being all predicted by the original LLM head. This decoding strategy serves as our baseline for efficient analysis.

### C.2  Tree Attention

**Tree construction.** Following the verification of token tree [50, 48], we merge the candidate tokens generated from multiple LLM heads to construct the tree. We employ a greedy search method from top to bottom to explore a layered graph and find the node sequences (paths) with maximum cumulative expectation. The expectation value is calculated by the estimated accuracy of each head. It starts from a root node and iteratively expands paths by selecting the neighbor with the highest expectation, computed as the product of node accuracies along the path. Neighbors are generated by either moving to the next node in the same layer or extending to the next layer, up to a specified maximum depth and child count per layer. We cache computed expectations to avoid redundant calculations and return a list of selected node sequences. We illustrate a tree structure example in Figure 11. We can observe that the token tree provides multiple token sequences (paths) for the following verification, thus improving the token acceptance rate.

**Tree decoding.** Upon the pre-defined token tree structure, we employ tree decoding to process the generated multiple predictions. First, we initialize the tree attention and indices given the tree paths (*cf.*, Figure 4). Once we generate multiple tokens' logits, we select the top-k candidates, which are assembled as input for the target models. By utilizing the tree attention, the target model yields the prediction of the original head for verification in parallel, finally accepting the candidates and starting the next iteration.

## D  Additional Experimental Results

We present the detailed training loss trends in Figure 12 for head warm-up and Figure 13 for full model tuning, with losses steadily decreasing and convergence to stability.

---

[8]https://docs.vllm.ai

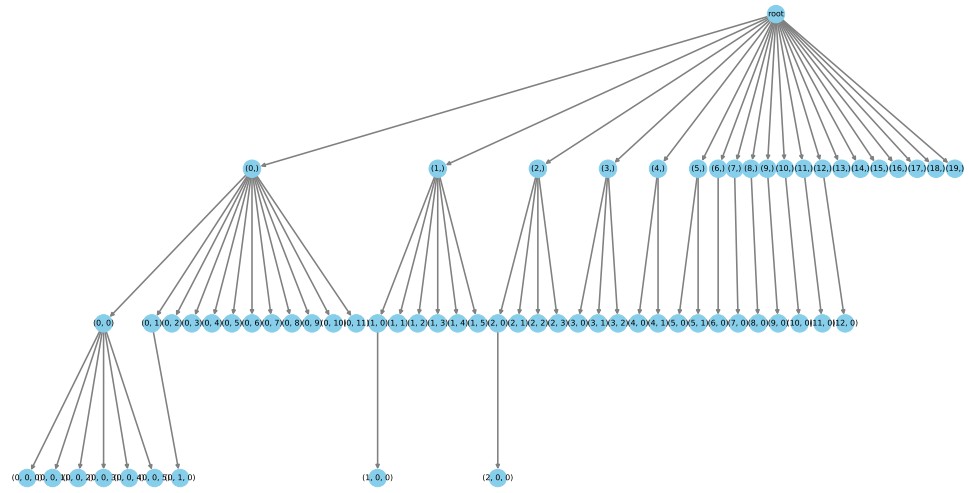

Figure 11: Token tree constructed according to the accuracy of heads. We use 3 heads for the Vicuna 7B model. The head accuracy is estimated when employing L-MTP decoding.

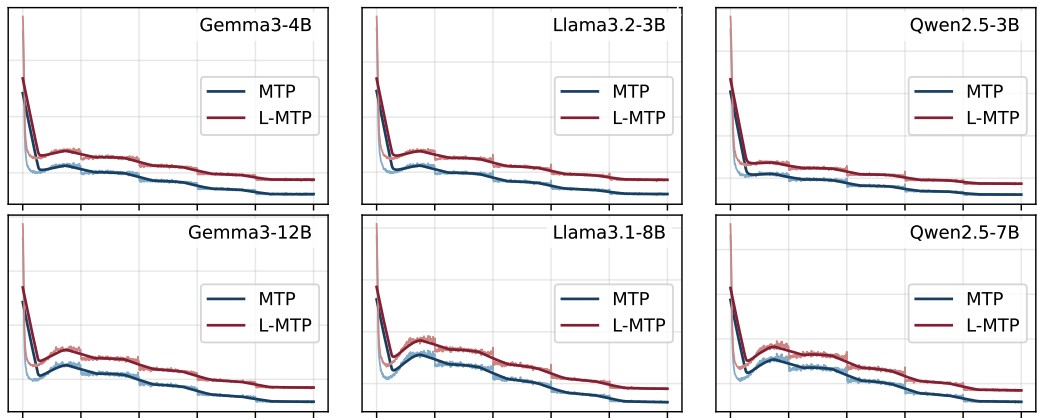

Figure 12: The illustration of the warm-up procedure of adapting multiple output heads to LLMs. The curves showcase the loss changes along with the head warm-up training for MTP and L-MTP.

# E    Broader Impact Statement

Leap multi-token prediction (L-MTP) redefines the traditional autoregressive paradigm by skipping intermediate tokens and predicting non-adjacent ones, mirroring human-like reasoning. This leap-based strategy broadens the contextual window, enhances inference efficiency, and reduces computational costs, making LLMs more suitable for complex decision-making tasks. Environmentally, L-MTP's efficient decoding lowers energy consumption and supports greener AI deployments. Its design also expands access to high-performance LLMs, reducing barriers for resource-constrained developers. Furthermore, L-MTP introduces new possibilities for non-sequential reasoning, which paves the way for more efficient and scalable language models.

# F    Reproducibility

We provide implementation details, including illustrative algorithm descriptions and pseudo-code. The source code is publicly released for reproducibility.

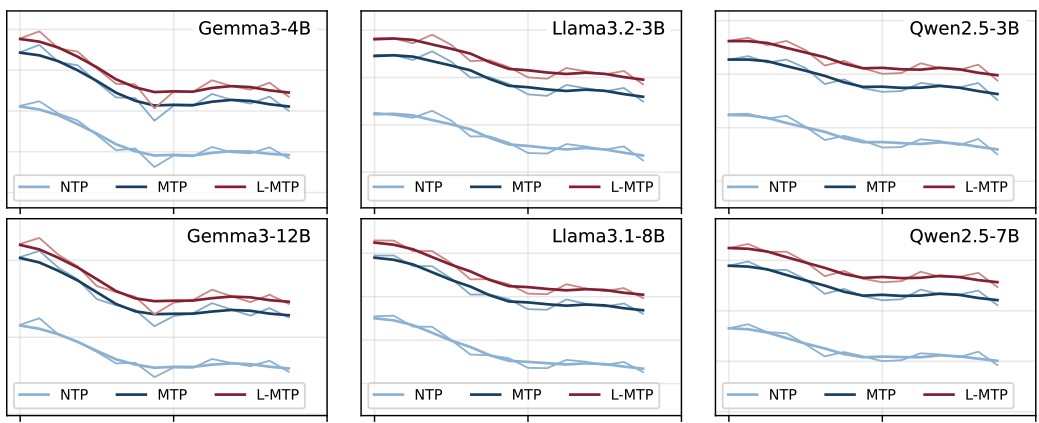

Figure 13: Full model fine-tuning with respect to different prediction paradigms. The curves showcase the loss changes along with the full model tuning for NTP, MTP, and L-MTP.

## G   Limitations

Modern large language models are rapidly scaling up, with recent models reaching tens or even hundreds of billions of parameters (*e.g.*, DeepSeek-R1-70B/671B [97] and Llama-3.1-405B [2]). Our experiments are conducted on models up to 12B due to computational constraints. Despite this limitation, the results effectively validate our core ideas. In future work, we plan to extend our method to larger models to further assess its scalability and effectiveness at greater capacity.

