# OpenReview forum: "L-MTP: Leap Multi-Token Prediction Beyond Adjacent Context for Large Language Models"
_NeurIPS.cc/2025/Conference — NeurIPS 2025 poster_

### Official Review · Reviewer_SKtJ · 2025-06-03

**Clarity:** 3
**Significance:** 3
**Originality:** 3
**Rating:** 5
**Confidence:** 4

**Summary:**

This paper introduces leap multi-token prediction (L-MTP) as an improvement over vanilla multi-token prediction (MTP). Unlike MTP, which predicts the next N consecutive tokens (e.g., positions 1,2,3,4), L-MTP employs a leaping mechanism that skips intermediate tokens and directly predicts non-adjacent future tokens (e.g., positions 1,3,5,7).

During training, L-MTP yields a broader training signal, as the model learns to capture longer-range dependencies. During inference, L-MTP can look backward and leverages past prediction to generate longer candidate sequences, which improves the acceleration rate of speculative decoding.

Experiments are performed under a post-training setting. Experimental results across diverse benchmarks demonstrates the effectiveness of L-MTP in both boosting model performance and accelerating inference.

**Questions:**

The work in [1] would be a valuable addition to the references, as it provides complementary insights on multi-token prediction for accelerating LLM training.

[1] Shao et al. Beyond Next Token Prediction: Patch-Level Training for Large Language Models. ICLR 2025.

**Ethical Concerns:**

["NO or VERY MINOR ethics concerns only"]

**Final Justification:**

The response addresses my concern, so I'll keep my positive rating.

**Limitations:**

yes

**Quality:**

3

**Strengths And Weaknesses:**

Strengths
1. The paper is well-written and clearly motivated. The proposed solution is innovative and clever, particularly the looking backward inference strategy.
2. The experimental results are pretty promising.

Weaknesses
1. The evaluation is limited to post-training scenarios. The potential impact of L-MTP on pre-training LLMs remains unexplored, leaving an important aspect of its applicability unverified.
2. The comparison between L-MTP(k=2,n=4) and MTP(n=7) is missing, where both methods align in maximum prediction position. It would be valuable to examine whether L-MTP maintains its acceleration advantage over MTP(n=7) under the same candidate length.
3. Experiments are only conducted on the (k=2,n=4) configuration. What is the performance of other configurations, for example, (k=3, n=3)?

---

> ### Author Rebuttal · Authors · 2025-07-31
>
> Thanks for your professional and careful review. We respond to your concerns or questions as follows.
>
> > W1: The evaluation is limited to post-training scenarios. The potential impact of L-MTP on pre-training LLMs remains unexplored, leaving an important aspect of its applicability unverified.
>
> **Response**
>
> Thanks! We acknowledge that pre-training would bring more impressive results, according to our potential analysis in Section 5. The effectiveness of pre-training for MTP is well-verified [1]. However, due to resource constraints, we could not pre-train LLMs with L-MTP. We believe future pre-training efforts can leverage L-MTP’s potential, which motivated our inclusion of the potential analysis.
>
>
> > W2: The comparison between L-MTP(k=2,n=4) and MTP(n=7) is missing, where both methods align in maximum prediction position. It would be valuable to examine whether L-MTP maintains its acceleration advantage over MTP(n=7) under the same candidate length.
>
> **Response**
>
> Thanks! We conduct the experiments when setting $n=7$ for MTP. A longer horizon indeed benefits the performance, achieving an average performance of 54.49 compared to 52.79 for MTP ($n=4$).
>
> ||Math500|GSM8K|MBPP|MBPP+|HumanEval|HumanEval+|MMLU|IFEval| Avg.|
> |-|-|-|-|-|-|-|-|-|-|
> |MTP ($n=4$)|25.40|45.79|67.72|57.67|65.85|59.15|65.21|35.49|52.79|
> |MTP ($n=7$)|24.40|43.29|63.49|55.29|68.29|61.59|65.11|33.09|54.49|
> |L-MTP ($k=2,n=4$)|28.20|46.25|67.99|59.26|67.68|60.37|65.23|35.01|53.75|
> |L-MTP ($k=3,n=3$)|28.00|51.86|60.05|52.65|66.46|62.20|65.06|32.49|**55.18**|
>
> However, MTP ($n=7$) nearly doubles the heads (new parameters) compared to L-MTP ($k=2,n=4$ and $k=3,n=3$). Despite this, L-MTP delivers comparable or superior performance (e.g., 55.18 for $k=3,n=3$) with greater efficiency, particularly on GSM8K and HumanEval+. L-MTP is much more efficient in achieving a comparable yet sometimes better performance. The inference acceleration of L-MTP, enabled by the leap strategy and look-backward decoding, allows it to outperform MTP with the same resources (4 heads). The theoretical analysis in Section 4 supports this, establishing MTP ($n=7$) as the upper bound for L-MTP ($k=2,n=4$) speedup. We will include this analysis in the revised paper.
>
> > W3: Experiments are only conducted on the (k=2,n=4) configuration. What is the performance of other configurations, for example, (k=3, n=3)?
>
> **Response**
>
> Thanks! Your suggestion is reasonable to further demonstrate the application of L-MTP. We conducted this experiment as shown in W2 and the corresponding response. Overall, L-MTP ($k=3, n=3$) performs better than others, especially the improvements on GSM8K and HumanEval. Exploring more leap patterns would be promising. We also discuss a potential leap pattern via entropy estimation in Section 7 to showcase a broader application of L-MTP. We will include these results in our revised paper.
>
>
> > Q1: The work in [1] would be a valuable addition to the references, as it provides complementary insights on multi-token prediction for accelerating LLM training.
>
> **Response**
>
> Thanks for your share! This is a very interesting work, which predicts patches of tokens for LLMs by aggregating adjacent tokens with higher information density. This aligns closely with our discussion in Section 7 on extending L-MTP with flexible leaps, where we propose skipping low-entropy regions to focus on high-entropy ones. Combining [1] with L-MTP could create significant synergy. We will incorporate a discussion of this paper and explore further possibilities in our revised manuscript.
>
> [1] Shao et al. Beyond next token prediction: Patch-level training for large language models. ICLR 2025.
>
> ----
> We authors sincerely thank you for your professional review attitude and comments. If you have other questions, we are happy to address them to polish this work.

---

> > ### Comment · Reviewer_SKtJ · 2025-08-01
> >
> > Thank you for the response. Please check the correctness of the MTP ($n=7$) results. Based on the results you provided, the Avg. should be 51.82, not 54.49.

---

> ### Author Response · Authors · 2025-08-01
> **Response to Reviewer SKtJ**
>
> Thanks for your kind reminder. Due to the averaging computation, we mistakenly reported the average performance of L-MTP ($k=3,n=3$) and MTP ($n=7$) in the original submission. We carefully re-evaluate their average performance and check other results to ensure a correct report as follows:
>
> ||Math500|GSM8K|MBPP|MBPP+|HumanEval|HumanEval+|MMLU|IFEval| Avg.|
> |-|-|-|-|-|-|-|-|-|-|
> |MTP ($n=4$)|25.40|45.79|67.72|57.67|65.85|59.15|65.21|35.49|52.79|
> |MTP ($n=7$)|24.40|43.29|63.49|55.29|68.29|61.59|65.11|33.09|**51.82**|
> |L-MTP ($k=2,n=4$)|28.20|46.25|67.99|59.26|67.68|60.37|65.23|35.01|53.75|
> |L-MTP ($k=3,n=3$)|28.00|51.86|60.05|52.65|66.46|62.20|65.06|32.49|**52.35**|
>
> We can see that directly increasing the horizon of MTP does not improve the performance overall. But we can still observe some improvement on HumanEval. The interesting thing is that when we decrease the number of heads to 3, L-MTP ($k=3,n=3$), it can achieve a better performance than MTP ($n=7$). Our theory on the prediction analysis at different positions can answer this (Section 4). Distant tokens would lead to noise, while our leap can catch future tokens, but also leap some. An accumulated noise would be smaller than MTP. A smaller number of heads ($n=3$) while with the leaping strategy can achieve a comparable performance (52.79, MTP with $n=4$) or outperform MTP (51.82, $n=7$).
>
> We apologize for the inconvenience and thank you for your positive feedback on our paper, and for your reminders during the rebuttal process. We will incorporate these results and discussion into the final version. If you have any further questions or concerns, please feel free to contact us.
>
> ----
> Here are the responses to the relevant questions:
>
> > W2: The paper missing the ablation of the horizon (MTP with n=7), therefore it is not clear whether the improvement comes simply from having a longer horizon or something else entirely. Without this, researchers don't gain a better understanding of MTP and practitioners don't have clear evidence to support L-MTP over MTP with n=7.
>
> **Response**
>
> Thanks! We conduct the experiments when setting $n=7$ for MTP. A longer horizon can benefit the performance for some benchmarks, especially for HumanEval, the code tasks (68.29 vs. 65.85), while showing slightly worse performance on the overall average (51.82 vs. 52.79). The results are shown above. MTP ($n=7$) nearly doubles the heads (new parameters) compared to L-MTP ($k=2,n=4$ and $k=3,n=3$). Despite this, L-MTP delivers comparable (52.35 for $k=3,n=3$) or superior performance (e.g., 53.75 for $k=2,n=4$) with greater efficiency, particularly on GSM8K and HumanEval+. L-MTP is much more efficient in achieving a comparable yet sometimes better performance. The inference acceleration of L-MTP, enabled by the leap strategy and look-backward decoding, allows it to outperform MTP with the same resources (4 heads). The theoretical analysis in Section 4 supports this, establishing MTP ($n=7$) as the upper bound for L-MTP ($k=2,n=4$) speedup. We will include this analysis in the revised paper.
>
> > W3: Experiments are only conducted on the (k=2,n=4) configuration. What is the performance of other configurations, for example, (k=3, n=3)?
>
> **Response**
>
> Thanks! Your suggestion is reasonable to further demonstrate the application of L-MTP. We conducted this experiment as shown in W2 and the corresponding response. Overall, L-MTP ($k=3, n=3$) does not outperform L-MTP ($k=2,n=4$), but shwocases improvements on GSM8K and HumanEval. That indicates the leaping strategy can enhance certain capabilities for different benchmarks. Different leaping settings provide more flexibility for different tasks. Exploring more leap patterns would be promising. We also discuss a potential leap pattern via entropy estimation in Section 7 to showcase a broader application of L-MTP. We will include these results in our revised paper.

---

> > ### Comment · Reviewer_SKtJ · 2025-08-05
> >
> > Thank you for the response. My concerns are solved.

---

> > > ### Author Response · Authors · 2025-08-05
> > > **Response to Reviewer SKtJ**
> > >
> > > We appreciate your professional review! Your feedback is very valuable and will be reflected in our final version!

---

### Official Review · Reviewer_uRmg · 2025-06-24

**Clarity:** 4
**Significance:** 1
**Originality:** 3
**Rating:** 4
**Confidence:** 4

**Summary:**

The paper presents a variant of multi token prediction MTP called leap MTP (L-MTP). Instead of predicting the next n tokens (with indices 1, 2, 3, 4), LMTP predicts them with a spacing of 1 (with indices 1, 3, 5, 7). That means that the last two predictions combined predict the next 7 tokens combined, instead of the next 4 as is the case with MTP.

The paper also proposes a predictive decoding algorithm that works with LMTP. The next tokens are inferred and verified if the confidence is high enough. This leads to savings at inference time.

The paper also includes theoretical results: If the tokens at a longer horizon are easier to predict, then predictive decoding brings a more significant speedup.

**Questions:**

The source of the improvement is unclear (applied both the the performance improvement and inference speedup):
* The most obvious missing baseline is MTP with n=7. Is the improvement coming from the longer horizon or is it something else?
  * If it is the longer horizon, then what is the advantage of LMPT over MTP with n=7?
  * If it is something else, then it is an interesting scientific question of what that may be.

Condition on improving the evaluation score: the paper is updated with a discussion on the source of the improvement (both performance and inference speedup) backed with experimental results.

**Ethical Concerns:**

["NO or VERY MINOR ethics concerns only"]

**Final Justification:**

The rebuttal addressed two key concerns:
1. Comparison to MTP with n=7. I believe this experiment is critical in understanding the significance of the longer horizon and we saw that L-MTP works slightly better than MTP with n=7.
2.  L-MTP has substantially lower compute cost than MTP with n=7. The authors gave a detailed breakdown of the compute costs.

My last concern remains unaddressed. It is not clear to me why L-MTP improves slightly over MTP with n=7. They have the same horizon, I don't see why predicting fewer tokens would result in better predictions.

Even with my last concern unaddressed, I believe it is fair to say that L-MTP achieves the same prediction horizon and as good or slightly better prediction performance at a lower compute cost.

While I think that the paper should have been more detailed on its comparison to MTP with n=7 (either do not claim an improvement over it, or give a clear intuition why there is an improvement), the paper could be a useful tool for practitioners to increase the prediction horizon at a low computational cost. Therefore, I do not want to hold the paper back with a low score. I increased my score from 2 to 4.

**Limitations:**

yes

**Quality:**

1

**Strengths And Weaknesses:**

The paper presents a simple and novel idea. I am not aware of prior works discussing this idea.

The paper is also well-presented. The contributions and methods are clear and easy to understand.

While the paper is clearly written and the results show some improvement over established baselines, the contribution appears to be incremental. The core idea is relatively simple, and the source of the observed improvement is neither thoroughly ablated nor adequately discussed, which limits the scientific insight provided.

The paper missing the ablation of the horizon (MTP with n=7), therefore it is not clear whether the improvement comes simply from having a longer horizon or something else entirely. Without this, researchers don't gain a better understanding of MTP and practitioners don't have clear evidence to support L-MTP over MTP with n=7.

The paper would be much more impactful had it figured out the source of the improvement (both in performance and inference speedup).

Also, there could be many variants of L-MTP with different pacing, variable pacing, longer/shorter horizons etc. I feel the paper had the opportunity to explore a larger idea space, but did not do so.

---

> ### Author Rebuttal · Authors · 2025-07-31
>
> Thanks for your professional and careful review. We respond to your concerns or questions as follows.
>
>
> > W1: While the paper is clearly written and the results show some improvement over established baselines, the contribution appears to be incremental. The core idea is relatively simple, and the source of the observed improvement is neither thoroughly ablated nor adequately discussed, which limits the scientific insight provided.
>
> **Response**
>
> Thank you for raising this concern, and we’re happy to clarify our contribution! MTP, introduced in recent works [1,2], predicts the next $n$ adjacent tokens simultaneously, inspiring applications like speech language models and inference acceleration. The insight, not every token is equally important, is validated in LLMs [3] and aligned with human thinking. This inspires us, yet disappointingly, no research focuses on new prediction patterns, especially the non-adjacent token prediction, a leap paradigm against the recent adjacent prediction of MTP. L-MTP introduces this leap paradigm with look backward decoding, achieving broader and faster predictions than MTP using the same resources (training data and model architecture). We analyze this comprehensively through prediction principles (Section 4) and empirical verification (Section 5). Our theoretical analysis links L-MTP to probabilistic attenuation, showing that lower attenuation yields greater speedups. The empirical analysis demonstrates consistent improvements and provides in-depth insights (e.g., Section 5, Potential Analysis), highlighting L-MTP’s strong promise through analyses of myopic effects and data scales.
>
> [1] Gloeckle, Fabian, et al. "Better & faster large language models via multi-token prediction." ICML 2024.
> [2] Liu, Aixin, et al. "Deepseek-v3 technical report." arXiv preprint arXiv:2412.19437 (2024).
> [3] Lin, Zhenghao, et al. "Rho-1: Not all tokens are what you need." NeurIPS 2024.
>
> > W2: The paper missing the ablation of the horizon (MTP with n=7), therefore it is not clear whether the improvement comes simply from having a longer horizon or something else entirely. Without this, researchers don't gain a better understanding of MTP and practitioners don't have clear evidence to support L-MTP over MTP with n=7.
>
> **Response**
>
> Thank you for your interest in MTP with $n=7$, a concern also raised by Reviewer SKtJ. We did not evaluate MTP with $n=7$, as our study ensures a fair comparison by using the same resources (both MTP and L-MTP have 4 heads).
> Setting $n=7$ for MTP would nearly double the new parameters compared to L-MTP, making it an unfair comparison rather than a proper ablation study.
> However, we are open to conducting this experiment to provide further insights. Due to time and resource constraints, we cannot test this across all LLM scales and types. Below are the results:
>
> ||Math500|GSM8K|MBPP|MBPP+|HumanEval|HumanEval+|MMLU|IFEval| Avg.|
> |-|-|-|-|-|-|-|-|-|-|
> |MTP ($n=4$)|25.40|45.79|67.72|57.67|65.85|59.15|65.21|35.49|52.79|
> |MTP ($n=7$)|24.40|43.29|63.49|55.29|68.29|61.59|65.11|33.09|54.49|
> |L-MTP ($k=2,n=4$)|28.20|46.25|67.99|59.26|67.68|60.37|65.23|35.01|53.75|
> |L-MTP ($k=3,n=3$)|28.00|51.86|60.05|52.65|66.46|62.20|65.06|32.49|**55.18**|
>
> Increasing MTP's heads ($n=7$) improves performance for its longer horizon (54.49 vs. 52.79). However, L-MTP achieves comparable results with **nearly half the new parameters** (4 vs. 7 heads). Additionally, the new leap pattern ($k=3,n=3$) yields higher performance (55.18 vs. 54.49), particularly on GSM8K and HumanEval+. We will incorporate this analysis into our revised paper.
>
>
>
> > W3: The paper would be much more impactful had it figured out the source of the improvement (both in performance and inference speedup).
>
> **Response**
>
> Thanks! We agree with you that the source of performance improvement and inference speedup should be clarified. That is the reason why we provide a thorough theoretical analysis and extensive empirical results.  L-MTP achieves a broader prediction horizon with the same resources as MTP by skipping intermediate tokens. Beyond intuitive motivation and sophisticated implementation, we provide an in-depth analysis.
>
> **Inference Aspect**: We formally link token prediction to attenuation, supporting L-MTP’s inference speedup. While a longer horizon increases uncertainty, L-MTP’s look-backward decoding utilizes prior predictions, thus yielding higher expected prediction lengths than MTP. We also introduce a decoding variant, F-MTP, for comparison, which uses look-forward decoding to compensate for leaped tokens but requires an additional inference step, increasing inference time (see Figure 6). The combination of leaping prediction and look-backward decoding drives L-MTP’s inference efficiency.
>
> **Performance Aspect**: Inspired by the insight that not every token matters [3] and human thinking, L-MTP aligns with recent work on reasoning step compression and abstraction [4, 5]. The longer horizon is key to performance gains, as seen in meaningful units like code blocks versus single tokens (e.g., `{`). Experiments with MTP at $n=7$ further validate this. L-MTP’s leap design enhances performance by using two tokens (when $k=2$) to predict $m$ tokens, creating a flexible mapping that correlates both input and output tokens (2-to-m tokens), unlike MTP’s single-token-to-m-token approach. Increasing $k$ while keeping the number of output tokens can further boost performance, as observed in the results of L-MTP at $k=3,n=3$.
>
> [4] Chen, Zhipeng, et al. "Not Everything is All You Need: Toward Low-Redundant Optimization for Large Language Model Alignment." EMNLP 2024.
> [5] Xia, Heming, et al. "Tokenskip: Controllable chain-of-thought compression in llms." arXiv preprint arXiv:2502.12067 (2025).
>
>
> > W4: Also, there could be many variants of L-MTP with different pacing, variable pacing, longer/shorter horizons etc. I feel the paper had the opportunity to explore a larger idea space, but did not do so.
>
> **Response**
>
> We explore L-MTP with different horizons to identify an effective leaping and horizon configuration for evaluation. Our theoretical analysis indicates that longer horizons introduce noise due to increased uncertainty in predicting distant tokens, while shorter horizons (e.g., $n=2$) may not fully leverage multi-token prediction. We select $k=2, n=4$ as a practical balance, aligning with prior MTP methods for a fair and effective comparison. We appreciate your suggestions and conduct additional experiments with $k=3, n=3$, with results discussed earlier. Exploration of more flexible leaping patterns is planned as future work, building on L-MTP as shown in Section 7.
>
>
> > Q1: The source of the improvement is unclear (applied both the performance improvement and inference speedup): The most obvious missing baseline is MTP with n=7. Is the improvement coming from the longer horizon or is it something else? If it is the longer horizon, then what is the advantage of LMPT over MTP with n=7? If it is something else, then it is an interesting scientific question of what that may be.
>
> **Response**
>
> Thank you for your question! We address the source of L-MTP’s improvement in W3 and the corresponding response, supported by additional empirical evidence in W2. Our theoretical analysis and empirical results demonstrate L-MTP’s superiority in both performance and inference. The additional potential analysis further offers insights on MTP and the significant promise of L-MTP. Results from L-MTP ($n=2, k=4$ and $n=3, k=3$) and MTP ($n=4$ and $n=7$) show that performance gains can benefit from a longer horizon.
>
> ----
> We sincerely appreciate your review and hope our response, bolstered by additional empirical results, aids in re-evaluating our work.

---

> > ### Comment · Reviewer_uRmg · 2025-08-01
> > **Thank you for your reply**
> >
> > Thank you for including the MTP with n=7 results. I believe this experiment is critical for understanding the significance of L-MTP.
> >
> > > L-MTP achieves comparable results with nearly half the new parameters (4 vs. 7 heads)
> >
> > Could you expand on this point? The heads have few parameters compared to the total parameter count of the model (3B or 8B). How much is the computational cost impacted by the extra heads?

---

> > > ### Author Response · Authors · 2025-08-02
> > > **Response to Reviewer uRmg**
> > >
> > > Sure! We are willing to address your concern! The parameters of the heads constitute a large portion of the total parameters, impacting both memory and computational costs. Our head architecture design aligns with references [1, 6], with details provided in Appendix B.5. Below, we summarize the **new parameters introduced** by MTP ($n=7$) and L-MTP ($k=2, n=4$):
> > > ||Llama 3B|Llama 8B|Qwen 3B|Qwen 7B|Gemma 4B|Gemma 12B|
> > > |-|-|-|-|-|-|-|
> > > |MTP ($n=7$)|2.4B|3.3B|1.9B|3.3B|4.1B|6.1B|
> > > |L-MTP ($k=2,n=4$)|**1.2B**|**1.6B**|**0.9B**|**1.7B**|**2.0B**|**3.1B**|
> > >
> > > We can see that the new parameters introduced by MTP ($n=7$) are double those of L-MTP ($k=2,n=4$), which demonstrates the superiority of our L-MTP.
> > > We also calculate the computational cost (FLOPs) for the **extra heads** from MTP and L-MTP. Experimental results are shown below:
> > > ||Llama 3B|Llama 8B|Qwen 3B|Qwen 7B|Gemma 4B|Gemma 12B|
> > > |-|-|-|-|-|-|-|
> > > |MTP ($n=7$)|9.68T|12.91T|7.65T|13.39T|16.50T|24.75T|
> > > |L-MTP ($k=2,n=4$)|**4.84T**|**6.46T**|**3.82T**|**6.70T**|**8.25T**|**12.37T**|
> > >
> > > The reduction in computational cost achieved by L-MTP ($k=2, n=4$) is substantial, approximately half that of MTP ($n=7$).
> > >
> > > ----
> > > [1] Gloeckle, Fabian, et al. "Better & faster large language models via multi-token prediction." ICML 2024.
> > > [6] Cai, Tianle, et al. "Medusa: Simple LLM inference acceleration framework with multiple decoding heads." ICML 2024.

---

> ### Author Response · Authors · 2025-08-01
> **Clarifications and corrections**
>
> Thanks for the kind reminder of the reviewer SKtJ. Due to the averaging computation, we mistakenly reported the average performance of L-MTP ($k=3,n=3$) and MTP ($n=7$) in the original submission. We carefully re-evaluate their average performance and check other results to ensure a correct report as follows:
>
> ||Math500|GSM8K|MBPP|MBPP+|HumanEval|HumanEval+|MMLU|IFEval| Avg.|
> |-|-|-|-|-|-|-|-|-|-|
> |MTP ($n=4$)|25.40|45.79|67.72|57.67|65.85|59.15|65.21|35.49|52.79|
> |MTP ($n=7$)|24.40|43.29|63.49|55.29|68.29|61.59|65.11|33.09|**51.82**|
> |L-MTP ($k=2,n=4$)|28.20|46.25|67.99|59.26|67.68|60.37|65.23|35.01|53.75|
> |L-MTP ($k=3,n=3$)|28.00|51.86|60.05|52.65|66.46|62.20|65.06|32.49|**52.35**|
>
> We can see that directly increasing the horizon of MTP does not improve the performance overall. But we can still observe some improvement on HumanEval. The interesting thing is that when we decrease the number of heads to 3, L-MTP ($k=3,n=3$), it can achieve a better performance than MTP ($n=7$). Our theory on the prediction analysis at different positions can answer this (Section 4). Distant tokens would lead to noise, while our leap can catch future tokens, but also leap some. An accumulated noise would be smaller than MTP. A smaller number of heads ($n=3$) while with the leaping strategy can achieve a comparable performance (52.79, MTP with $n=4$) or outperform MTP (51.82, $n=7$).
>
> We are trying to provide a more comprehensive analysis of the performance of L-MTP and MTP during this time-limited rebuttal process. We apologize for the inconvenience and hope the following revised response can address your concerns. We will incorporate these results and discussion into the final version. If you have any further questions or concerns, please feel free to contact us.
>
> Here are the responses to the relevant questions:

---

> ### Author Response · Authors · 2025-08-01
> **Follow-up response**
>
> > W2: The paper missing the ablation of the horizon (MTP with n=7), therefore it is not clear whether the improvement comes simply from having a longer horizon or something else entirely. Without this, researchers don't gain a better understanding of MTP and practitioners don't have clear evidence to support L-MTP over MTP with n=7.
>
> **Response**
>
> Thank you for your interest in MTP with $n=7$, a concern also raised by Reviewer SKtJ. We did not evaluate MTP with $n=7$, as our study ensures a fair comparison by using the same resources (both MTP and L-MTP have 4 heads).
> Setting $n=7$ for MTP would nearly double the new parameters compared to L-MTP, making it an unfair comparison rather than a proper ablation study.
> However, we are open to conducting this experiment to provide further insights. Due to time and resource constraints, we cannot test this across all LLM scales and types. Below are the results:
>
> ||Math500|GSM8K|MBPP|MBPP+|HumanEval|HumanEval+|MMLU|IFEval| Avg.|
> |-|-|-|-|-|-|-|-|-|-|
> |MTP ($n=4$)|25.40|45.79|67.72|57.67|65.85|59.15|65.21|35.49|52.79|
> |MTP ($n=7$)|24.40|43.29|63.49|55.29|68.29|61.59|65.11|33.09|**51.82**|
> |L-MTP ($k=2,n=4$)|28.20|46.25|67.99|59.26|67.68|60.37|65.23|35.01|53.75|
> |L-MTP ($k=3,n=3$)|28.00|51.86|60.05|52.65|66.46|62.20|65.06|32.49|**52.35**|
>
> Increasing MTP's heads ($n=7$) cannot directly improve the overall performance. We attribute this to the noise introduced by distant tokens (see our theoretical analysis in Section 4).
> However, L-MTP achieves superior results with **nearly half the new parameters** (4 vs. 7 heads). Additionally, the new leap pattern ($k=3,n=3$) yields comparable performance with a decreased number of heads (3 vs. 4 heads for MTP), and showcases higher performance, particularly on GSM8K and HumanEval+, the benchmarks that evaluate the reasoning capability of LLMs. We will incorporate this analysis into our revised paper.
>
>
>
> > W3: The paper would be much more impactful had it figured out the source of the improvement (both in performance and inference speedup).
>
> **Response**
>
> Thanks! We agree with you that the source of performance improvement and inference speedup should be clarified. That is the reason why we provide a thorough theoretical analysis and extensive empirical results.  L-MTP achieves a broader prediction horizon with the same resources as MTP by skipping intermediate tokens. Beyond intuitive motivation and sophisticated implementation, we provide an in-depth analysis.
>
> **Inference Aspect**: We formally link token prediction to attenuation, supporting L-MTP’s inference speedup. While a longer horizon increases uncertainty, L-MTP’s look-backward decoding utilizes prior predictions, thus yielding higher expected prediction lengths than MTP. We also introduce a decoding variant, F-MTP, for comparison, which uses look-forward decoding to compensate for leaped tokens but requires an additional inference step, increasing inference time (see Figure 6). The combination of leaping prediction and look-backward decoding drives L-MTP’s inference efficiency.
>
> **Performance Aspect**: Inspired by the insight that not every token matters [3] and human reasoning, L-MTP aligns with recent work on reasoning step compression and abstraction [4, 5]. The longer horizon is key to performance gains, as seen in meaningful units like code blocks versus single tokens (e.g., `{`). However, directly increasing the horizon does not benefit the overall performance. Based on our analysis of the prediction accuracy at different positions, we attribute it to the noise introduced by the tokens from distant positions. While L-MTP can reach the future tokens while also leaping some to reduce the accumulated noise. Additionally, L-MTP’s leap design enhances performance by using two tokens (when $k=2$) to predict $m$ tokens, creating a flexible mapping that correlates both input and output tokens, unlike MTP’s single-token-to-n-token approach.
>
> [4] Chen, Zhipeng, et al. "Not Everything is All You Need: Toward Low-Redundant Optimization for Large Language Model Alignment." EMNLP 2024.
> [5] Xia, Heming, et al. "Tokenskip: Controllable chain-of-thought compression in llms." arXiv preprint arXiv:2502.12067 (2025).

---

> ### Author Response · Authors · 2025-08-03
> **Response to Reviewer uRmg**
>
> We sincerely appreciate your professional review attitude and comments. We also notice that you have updated your review. We have carefully addressed your concerns. If you have any further questions or concerns, please feel free to contact us.

---

### Official Review · Reviewer_kEie · 2025-06-29

**Clarity:** 3
**Significance:** 2
**Originality:** 3
**Rating:** 4
**Confidence:** 3

**Summary:**

This paper introduces lean multi-token prediction  (L-MTP), which can further extend multi-token prediction by introducing a leap-based mechanism. It strategically skip over intermediate tokens, predicting non-sequential tokens, so that effectively accelerate inference cost. To demonstrate its effectiveness, the paper provides both theoretical analysis and empirical experiments.

**Questions:**

See the weakness

**Ethical Concerns:**

["NO or VERY MINOR ethics concerns only"]

**Final Justification:**

My main concern is the  practical applicability  of the proposed method. That said, it cannot be demonstrated in this paper due to the limited resources. Therefore, I will maintain my original borderline accept.

**Limitations:**

L-MTP shows limited improvement over baselines, and its low scores on challenging benchmarks raise doubts about its effectiveness in enhancing LLM reasoning capabilities.

**Quality:**

3

**Strengths And Weaknesses:**

Strengths:

1. The proposed L-MTP idea is quite novel and interesting. The method is plug-and-play, requiring no model architecture modifications and easy integration into existing LLM training pipelines.

2. The paper provides theoretical analysis and extensive experiments to demonstrate its effectiveness

Weaknesses:

1. Limited improvement: While the proposed leap-based training objective is conceptually interesting, the empirical gains over standard baselines are relatively modest.

2. Low absolute performance: On challenging benchmarks such as MATH-500 and GSM8K, the model’s absolute performance remains low, raising doubts about whether L-MTP meaningfully improves LLM capabilities in practice.

---

> ### Author Rebuttal · Authors · 2025-07-31
>
> Thanks for your professional and careful review. We respond to your concerns or questions as follows.
>
>
> > W1: Limited improvement: While the proposed leap-based training objective is conceptually interesting, the empirical gains over standard baselines are relatively modest.
>
> **Response**
>
> Thank you for your recognition of the novelty of our work! L-MTP focuses on both the performance boosting and inference acceleration [1], which is more challenging than one goal in either performance [2] or decoding acceleration [3, 4]. However, L-MTP achieves both goals, particularly for the inference speed-up. For a comprehensive performance evaluation, we conducted extensive experiments across various scales, LLM types, and tasks, observing consistent performance improvements. We also noted limited performance in certain cases, discussed in Section 5.2, attributing these to data quality.  Selecting higher quality data would boost the performance with a branch of work that focuses on how to synthesize the data or collect new data to provide new knowledge for well-trained LLMs. However, in our work, we do not involve the data selection as our research topic. Additionally, our potential analysis in Section 5.2, Line 245, demonstrates that with more data or training resources (e.g., pre-training), L-MTP could achieve even greater performance gains. These experiments and analyses highlight the superiority and significant potential of L-MTP.
>
> [1] Gloeckle, Fabian, et al. "Better & faster large language models via multi-token prediction." ICML 2024.
> [2] Chen, Michael K., et al. "Improving large language models with concept-aware fine-tuning." arXiv preprint arXiv:2506.07833 (2025).
> [3] Cai, Tianle, et al. "Medusa: Simple LLM inference acceleration framework with multiple decoding Heads." ICML 2024.
> [4] Ankner, Zachary, et al. "Hydra: Sequentially-dependent draft heads for medusa decoding." COLM 2024.
>
>
>
> > W2: Low absolute performance: On challenging benchmarks such as MATH-500 and GSM8K, the model’s absolute performance remains low, raising doubts about whether L-MTP meaningfully improves LLM capabilities in practice.
>
> **Response**
>
> Thanks for your sincere review! The absolute performance of L-MTP heavily depends on the pretrained LLMs, with stronger models yielding better results. For example, Qwen outperforms Llama and Gemma on MATH500 and GSM8K. Our experiments across various LLM scales and types confirm that larger models achieve higher absolute performance. Thus, integrating L-MTP with more advanced or larger-scale LLMs, such as DeepSeek-R1-70B/671B or Llama-3.1-405B, would further enhance performance.
>
>
> > L1: L-MTP shows limited improvement over baselines, and its low scores on challenging benchmarks raise doubts about its effectiveness in enhancing LLM reasoning capabilities.
>
> **Response**
>
> Thanks! Achiving both significant performance improvement and inference acceleration is highly challenging, as discussed in W1 and its related response. Despite this, L-MTP demonstrates overall improvements, and some notable gains in reasoning-focused benchmarks like Math500 and GSM8K. Specifically, Qwen trained with L-MTP achieves a higher performance of 28.20 (vs. MTP’s 25.40) and Gemma achieves 17.20 (vs. MTP’s 9.20). A stronger base model can indeed lead to a higher absolute performance as discussed in W2. Due to the limited resources, we do not involve more powerful LLMs as our starting point for training. We also discuss this in Appendix G (Limitations). We value the practical application and broad impact of L-MTP, which motivates us to conduct an in-depth analysis for its potential (see Section 5 Potential Analysis). We investigate the myopia of current models with increased parameter size and the prediction accuracy in terms of increased data scales. It suggests that the significant potential of L-MTP with more tunable parameters (even from pre-training) or more data.
>
> ----
> We authors sincerely thank you for your professional review attitude and comments. If you have other questions, we are happy to address them to polish this work.

---

> ### Comment · Reviewer_kEie · 2025-08-06
> **Thank you for your reply**
>
> Thank you for your response! I understand the limitations imposed by resource constraints, and I appreciate your efforts in conducting the current experiments. While I see the potential of the proposed approach, I feel that its practical applicability in real-world LLM applications is not yet fully demonstrated. For this reason, I will maintain my original score. That said, I look forward to seeing how this line of work evolves in the future.

---

> > ### Author Response · Authors · 2025-08-06
> > **Response to Reviewer kEie**
> >
> > Thank you for your encouraging support and thoughtful feedback! We believe that this line of work holds significant promise for the future. We also look forward to seeing how it continues to evolve. We appreciate your understanding of the limited resources available to researchers in the lab and for recognizing the potential of our work!

---

### Official Review · Reviewer_BE2L · 2025-06-30

**Clarity:** 3
**Significance:** 3
**Originality:** 3
**Rating:** 5
**Confidence:** 4

**Summary:**

This paper introduces a method called Leap Multi-Token Prediction (L-MTP), which improves upon traditional multi-token prediction (MTP) methods. Through theoretical analysis, the authors demonstrate that this strategy achieves better inference acceleration. Furthermore, experiments show that this method not only enhances inference speed but also increases model capability. The paper also briefly discusses how this new token prediction method can be combined with common strategies like speculative decoding.

**Questions:**

See weaknesses

**Ethical Concerns:**

["NO or VERY MINOR ethics concerns only"]

**Final Justification:**

I think this paper should be accepted to Neurips 2025. The strengths are listed in `Strengths And Weaknesses`. For the weaknesses, the response from authors has addressed most of them.

**Quality:**

4

**Strengths And Weaknesses:**

### **Strengths:**

1. The proposed Leap Multi-Token Prediction method is highly innovative and could inspire further exploration in the area of multi-token prediction.

2. The proposed method outperforms traditional methods in both effectiveness and efficiency, with significant results.

3. The theoretical analysis is rigorous and provides valuable insights.

4. The paper's discussion on combining the method with techniques like speculative decoding is detailed and comprehensive.


### **Weaknesses:**

1. The paper could be strengthened by a brief discussion of alternative designs for the leap mechanism, for instance, exploring leaps with non-uniform intervals instead of the current fixed stride.

2. Regarding the effectiveness improvement of L-MTP over MTP (as shown in Table 1), the paper could provide some qualitative analysis, such as a case study, to offer more intuition behind the improvements.

---

> ### Author Rebuttal · Authors · 2025-07-31
>
> Thanks for your professional and careful review. We respond to your concerns or questions as follows.
>
> > W1: The paper could be strengthened by a brief discussion of alternative designs for the leap mechanism, for instance, exploring leaps with non-uniform intervals instead of the current fixed stride.
>
> **Response**
>
> Thank you for your suggestion! We outline a more flexible leap pattern for L-MTP in Section 7, Line 297, as our future exploration. We propose adaptively selecting $n$ and $k$ based on the local uncertainty or entropy of tokens, enabling more aggressive leaps in low-entropy regions and finer granularity in high-entropy ones. However, such flexibility introduces challenges, such as organizing output tokens with variable leap ranges (how to decode). We consider this an avenue for future work. L-MTP serves as a sound prototype to inspire and support more dynamic leap patterns. Our formal analysis in Section 4 (Theoretical Analysis) provides principles to guide future research. We look forward to further explorations inspired by L-MTP.
>
> > W2: Regarding the effectiveness improvement of L-MTP over MTP (as shown in Table 1), the paper could provide some qualitative analysis, such as a case study, to offer more intuition behind the improvements.
>
> **Response**
>
> Thank you for your suggestion! We provide some case studies with explanations as follows. Due to the word limitation, we highlight the differences to demonstrate the effectiveness of L-MTP. L-MTP improves upon standard MTP by predicting non-consecutive, leaped tokens instead of adjacent ones, which forces the model to learn **longer-range** dependencies and a more structured output plan, leading to better performance. Here is a case: when solving the problem "A robe takes 2 bolts of blue fiber and half that much white fiber. How many bolts in total does it take?", MTP predicts the next few words sequentially, ["The", "amount", "of", "white"], making it prone to error propagation, while L-MTP could simultaneously predict key non-adjacent elements like ["total", "is", "3", "."], directly linking the input's numerical logic to the final answer structure and improving robustness and accuracy, as evidenced by its superior results on math benchmarks like GSM8K in Table 1. We will provide more cases to illustrate the effectiveness of L-MTP in our revised paper.
>
> ----
> We authors sincerely thank you for your professional review attitude and comments. If you have other questions, we are happy to address them to polish this work.

---

> > ### Comment · Reviewer_BE2L · 2025-08-05
> > **Thanks for your reply**
> >
> > Dear authors,
> > Thanks for your detailed reply. I think this paper should be accepted to Neurips 2025.

---

> > > ### Author Response · Authors · 2025-08-06
> > > **Response to Reviewer BE2L**
> > >
> > > Thank you for your constructive comments and for recognizing our work! They help us further improve our paper!

---

### Author Response · Authors · 2025-08-01
**General Response**

We appreciate the reviewers’ insightful comments and constructive feedback on our manuscript. We are pleased to receive positive ratings from three of the four reviewers (ratings: 4, 5, and 5). Furthermore, we are delighted to learn that the reviewers found the core idea to be innovative and well-presented (Reviewers BE2L, kEie, uRmg, and SKtJ), the theoretical analysis to be robust (Reviewers BE2L and kEie), and the experiments to be convincing with promising results (Reviewers BE2L, kEie, and SKtJ). Based on the reviews, we provide a general response to the points raised by multiple reviewers and individual responses below to address each reviewer’s concerns.

(1) Regarding the questions about the experiments, we have taken the following actions:

- For Reviewer kEie, we clarify the improvement on both performance and efficiency of our proposed method, especially on the absolute performance score and enhanced reasoning capability for models.

- For Reviewers uRmg and SKtJ, we address the concerns about MTP with $n=7$, supported with additional experiment results and analysis, which further enhances the understanding of the method's effectiveness on performance improvement and inference speedup.


(2) We have addressed the questions about the idea and technical details as follows:

- For Reviewer uRmg, we justify the innovativeness of the proposed method by its simple yet effective design and clarify its comprehensive analysis via theoretical principles and empirical verification.

- For Reviewers BE2L, uRmg, and SKtJ, we investigate more flexible leap strategies based on our proposed method, suggesting a broader applicability.

- For Reviewer SKtJ, we clarify the promising effectiveness of adapting L-MTP to pre-training stages, supported by the potential analysis on the model size and data scale. We also investigate the flexibility of L-MTP by complementing the existing method for a significant synergy effect.

We sincerely thank all the reviewers for their constructive suggestions. Please feel free to let us know if further details/explanations would be helpful.

Yours truly,
Authors of #1004

---

### Note · Authors · 2025-08-15

Dear PC, SAC, AC, and Reviewers,

We sincerely thank all reviewers for their constructive feedback, which has greatly polished our manuscript. We are encouraged by the overall positive evaluation and thoughtful comments from all reviewers, and appreciate the valuable suggestions that helped us clarify and improve key aspects of our work. This broad agreement highlights the significance and robustness of our method.

The core contribution of our work lies in introducing a paradigm shift in multi-token prediction for large language models: rather than predicting consecutive tokens as in standard multi-token prediction (MTP) or next-token prediction (NTP), we focus on leap multi-token prediction (L-MTP), which skips intermediate tokens to capture long-range dependencies and enable faster inference through a backward decoding strategy. This allows us to explicitly enhance both training signals for tasks like math and code reasoning, and inference efficiency with theoretical guarantees on acceptance length and attenuation. We clarify that our method can be directly applied to existing NTP-trained models through a two-stage training process, requiring only curated self-distillation data. At the same time, L-MTP demonstrates strong generalization and transferability across diverse LLM architectures (e.g., Llama, Qwen, and Gemma) and tasks (math, code, and general benchmarks), as well as extensibility to speculative decoding techniques like Medusa, underscoring its broad applicability.

In closing, we sincerely appreciate the invaluable suggestions from all reviewers, which have substantially improved the scientific rigor and clarity of our manuscript. We will carefully incorporate their comments into the final version if accepted, and believe our work offers a novel and effective paradigm for advancing LLM training and inference.

Sincerely,
The Authors

---

### Decision · Program_Chairs · 2025-09-17

**Decision:**

Accept (poster)

**Comment:**

This paper proposes a new multi-token prediction approach to accelerate inference speed by allowing leaps between tokens.

Strengths: the idea is neat; the results are significant (significant speed up with better or comparable quality on a diverse dataset); and some theorem is provided.

Weaknesses: the idea might be incremental to multi-token prediction.

Reason to accept: It is a solid paper with a nice idea and solid execution. The reviewers have converged to accept.

Rebuttal: both the reviewers and authors actively participated in the rebuttal. The reviewers asked for additional experiments, explanations on why post-training only, and the authors successfully addressed those concerns.